# Increased Importance of Aerosol-Cloud Interaction for Surface PM$_{2.5}$ Pollution Relative to Aerosol-Radiation Interaction in China with the Anthropogenic Emission Reduction

Da Gao[1, 2], Bin Zhao[1, 2, *], Shuxiao Wang[1, 2], Yuan Wang[3], Brian Gaudet[4], Yun Zhu[5], Xiaochun Wang[1, 2], Jiewen Shen[1, 2], Shengyue Li[1, 2], Yicong He[1, 2], Dejia Yin[1, 2], Zhaoxin Dong[1, 2]

[1]State Key Joint Laboratory of Environment Simulation and Pollution Control, School of Environment, Tsinghua University, 100084 Beijing, China

[2]State Environmental Protection Key Laboratory of Sources and Control of Air Pollution Complex, Beijing, 100084, China

[3]Department of Earth, Atmospheric, and Planetary Sciences, Purdue University, West Lafayette, IN, USA,

[4]Pacific Northwest National Laboratory, Richland, Washington, USA

[5]Guangdong Provincial Key Laboratory of Atmospheric Environment and Pollution Control, College of Environment and Energy, South China University of Technology, Guangzhou Higher Education Mega Center, Guangzhou, 510006, China

*Correspondence to: Bin Zhao (bzhao@mail.tsinghua.edu.cn)

**Abstract:** Surface fine particulate matter (PM$_{2.5}$) pollution can be enhanced by

feedback processes induced by aerosol-radiation interactions (ARI) and aerosol-cloud interactions (ACI). Many previous studies have reported enhanced $PM_{2.5}$ concentration induced by ARI and ACI for episodic events in China. However, few studies have examined the changes in the ARI- and ACI-induced $PM_{2.5}$ enhancements over a long period, though the anthropogenic emissions have changed substantially in the last decade. In this study, we quantify the ARI- and ACI-induced $PM_{2.5}$ changes for 2013–2021 under different meteorology and emission scenarios using the Weather Research and Forecasting model with Chemistry (WRF-Chem) and investigate the driving factors for the changes. Our results show that in January 2013, when China suffered from the worst $PM_{2.5}$ pollution, the $PM_{2.5}$ enhancement induced by ARI in eastern China (5.59 $\mu g\ m^{-3}$) is larger than that induced by ACI (3.96 $\mu g\ m^{-3}$). However, the ACI-induced $PM_{2.5}$ enhancement shows a significantly smaller decrease ratio (51%) than the ARI-induced enhancement (75%) for 2013–2021, making ACI more important for enhancing $PM_{2.5}$ concentrations in January 2021. Our analyses suggest that the anthropogenic emission reduction plays a key role in this shift. Owing to only anthropogenic emission reduction, the ACI-induced $PM_{2.5}$ enhancement decreases by 43% in January, lower than the decrease ratio of the ARI-induced enhancement (57%). The relative change in ARI- and ACI-induced $PM_{2.5}$ enhancement in July is similar to the pattern observed in January caused by anthropogenic emission reduction. The primary reason for this phenomenon is that the decrease of ambient $PM_{2.5}$ for 2013–2021 causes a disproportionately

small decrease of liquid water path (LWP) and increase of cloud effective radius

(Re) under the condition of high $PM_{2.5}$ concentration. Therefore, the surface solar

radiation attenuation (and hence boundary layer height reduction) caused by ACI

decreases slower than that caused by ARI. Moreover, the lower decrease ratio of

the ACI-induced $PM_{2.5}$ enhancement is dominated by the lower decrease ratio of

ACI-induced secondary $PM_{2.5}$ component enhancement, which is additionally

caused by smaller decrease ratio of the air temperature reduction and relative

humidity (RH) increase. Our findings indicate that, with the decrease of ambient

$PM_{2.5}$, the ACI-induced $PM_{2.5}$ enhancement inevitably becomes more important.

This needs to be considered in the formulation of control policies to meet the

national $PM_{2.5}$ air quality standard.

## 1. Introduction

Aerosol-radiation interaction (ARI) and aerosol-cloud interaction (ACI) are

important ways for aerosols to influence the climate (Rosenfeld et al., 2014;

Seinfeld et al., 2016; Liu et al., 2018; Bellouin et al., 2020; Forster et al., 2021).

The ARI represents the direct scattering and absorption of solar and infrared

radiation by atmospheric aerosols; the ACI denotes the modification effects on

the lifetime, physical and optical properties of clouds by atmospheric aerosols.

Previous studies have documented that both ARI and ACI have important

contributions to inhibiting the planetary boundary layer height (PBLH), cooling

the near-surface air temperature, and increasing the relative humidity (RH) (Wang

et al., 2014; Ding et al., 2016; Liu et al., 2018). Moreover, ACI has extra

contributions to changing precipitation and cloud chemistry (Zhao et al., 2017; Zhang et al., 2018). These feedbacks and changes are mostly conducive to increasing the haze severity (Wang et al., 2015; Zhang et al., 2018; Liu et al., 2018; Zhou et al., 2019; Zhang et al., 2020; Xiong et al., 2022; Lin et al., 2022). So far, numerous studies have evaluated the fine particulate matter ($PM_{2.5}$) enhancements caused by the decreases of downward shortwave radiation at the surface (SWDOWN), PBLH, near-surface air temperature and precipitation, and by the increase of RH, especially during the severe $PM_{2.5}$ pollution in China (Le et al., 2020). Zhang et al. (2015) and Zhang et al. (2018) quantified that the ARI caused the $PM_{2.5}$ increase by 8.3 $\mu g\, m^{-3}$ in 2013 and 4.0 $\mu g\, m^{-3}$ in 2014. However, both positive and negative contributions of ACI to the $PM_{2.5}$ have been revealed (Forkel et al., 2012; 2015; Kong et al., 2015; Zhang et al., 2015; Zhang et al., 2018). Zhao et al. (2017) pointed out that the negative contribution of ACI shown in some studies (Gustafson et al., 2007; Gong et al., 2015) is due to the relatively high prescribed values of cloud droplet number concentration (CDNC) or cloud condensation nuclei (CCN), which could not represent a rather clean condition. Besides, there might be a discrepancy between the enhancements induced by ARI and ACI for primary and secondary $PM_{2.5}$ components. The primary $PM_{2.5}$ components are mainly influenced by physical transport, while the secondary $PM_{2.5}$ components are also affected by chemical formation and decomposition. The lower air temperature and higher RH can help to condense gas precursors into secondary aerosol particles (Donahue et al., 2012) and strengthen aqueous

and heterogeneous reactions (Liu et al., 2018). On the contrary, Wu et al. (2020) pointed out that the ARI may also suppress the formation of secondary aerosol because the atmospheric oxidizing capacity and photolysis rate can be changed during the scattering and absorbing of solar radiation. Therefore, not all changes of meteorological factors are conducive to the increase of secondary $PM_{2.5}$, and these positive and negative contributions would influence the variations of primary and secondary $PM_{2.5}$ components. In a word, although the ARI and ACI processes mostly lead to a net $PM_{2.5}$ increase, the relative increasing rates of different aerosol components are fairly complex due to various physical and chemical processes.

In recent years, the Chinese government has successively proclaimed the policies of "Air pollution prevention and control action plan" and "Three-year action plan to win the blue sky defense war", including the promotion of ultra-low emission technologies in industrial sectors, the implementation of traffic restriction policies, and the transition from coal to gas in residential cooking. As a result, the annually averaged $PM_{2.5}$ concentrations in Beijing-Tianjin-Hebei region, Yangtze River Delta (YRD) and Pearl River Delta have been reduced by 39.6%, 34.2%, and 27.7% from 2013 to 2017, respectively (Wang et al., 2017; Ding et al., 2019a). Meanwhile, sulfate and organic components have respectively decreased by 76% and 70 % in the North China Plain (NCP) (Wang et al., 2019). Considering the sharp anthropogenic emission reduction and $PM_{2.5}$ concentration decrease, Moch et al. (2022) found that the decrease in mean $PM_{2.5}$ concentration

from the winter months of 2012–2013 to the winter months of 2016–2017 in China weakened the cloud–snowfall–albedo feedback induced by the aerosol semi-direct effect. For air quality, Zhang et al. (2022) found that the decrease in black carbon from 2013 to 2017 in China reduced the enhanced $PM_{2.5}$ concentration induced by the ARI by 1.8 μg m$^{-3}$ in January and 0.3 μg m$^{-3}$ in July.

However, none of the previous studies have systematically evaluated the changes in enhanced $PM_{2.5}$ concentrations through ARI and ACI in China at the long-term scale. Besides, the driving force and physical mechanisms for the changes are also yet to be explored. In this study, we try to investigate the enhanced $PM_{2.5}$ concentrations induced by ARI and ACI in 2013 over China, the impact of the changes in the meteorological background and anthropogenic emission from 2013 to 2021 on ARI- and ACI-induced $PM_{2.5}$ enhancements and its components. Furthermore, the causes of $PM_{2.5}$ enhancement changes are analyzed.

## 2. Model and experimental design

## 2.1 Model configuration

The Weather Research and Forecasting model with Chemistry (WRF-Chem) version 4.2 has been used in this study. The model domain covers the whole land area of China with a horizontal resolution of 27 km × 27 km. There are 24 vertical layers from surface to 50 hPa, with denser layers in the planetary boundary layer

(PBL). Major physical options used in the model include the Morrison double-
moment scheme (Morrison et al., 2009), the Rapid Radiative Transfer Model for
GCMs (RRTMG) shortwave and longwave radiative transfer schemes (Iacono et
al., 2008), the Eta similarity surface-layer scheme (Janjic et al., 1994), the Noah
land-surface model with multiple parameterization options (Niu et al., 2011), the
Bougeault and Lacarrere PBL scheme (Bougeault et al., 1989), and the Grell-
Freitas ensemble cumulus scheme (Grell et al., 2014). For chemistry, we employ
the SAPRC-99 (Statewide Air Pollution Research Center mechanism, version
1999) as the gas-phase chemistry mechanism (Carter et al., 2000). The aerosol
module used in the study is the Model for Simulating Aerosol Interactions and
Chemistry (MOSAIC) (Zaveri et al., 2008), which includes all major aerosol
processes and represents the aerosol size distribution with 8 size bins. The
MOSAIC also incorporates the one-dimensional Volatility Basis Set (VBS)
framework that improves the simulation of secondary organic aerosol
(Shrivastava et al., 2011). Rates for photolytic reactions are calculated using the
Fast-J photolysis rate scheme (Wild et al., 2000). Additionally, we noted the poor
ability of nitrate simulation in the WRF-Chem model. We improved the nitrate
simulation by addressing the HONO underestimation in the model (Wang et al.,
2015; Xue et al., 2020). More detailed information can be found in Section 1 in
the Supplementary Information. The meteorological initial and boundary
conditions are derived from the National Centers for Environmental Prediction
Final Analysis reanalysis data with resolutions of $1.0° \times 1.0°$ and 6 h

(*http://rda.ucar.edu/datasets/ds083.2/*). The chemical initial and boundary
conditions are acquired from the simulation results of the National Center for
Atmospheric Research's Community Atmosphere Model with Chemistry (CAM-
Chem, before 2020, *https://www.acom.ucar.edu/cam-chem/cam-chem.shtml*) and
the Whole Atmosphere Community Climate Model (WACCM, after 2020,
*https://www.acom.ucar.edu/waccm/download.shtml*) with resolutions of 0.94° ×
1.25° and 6 h.

The anthropogenic emission data in China for 2013-2021 are obtained from
the ABaCAS-EI (Air Benefit and Cost and Attainment Assessment System-
Emission Inventory) developed by Tsinghua University (Li et al., 2023). Specific
emissions of $SO_2$, $NO_x$ (NO and $NO_2$), $NH_3$, $PM_{2.5}$ and VOCs in 2013 and 2021
are presented in Table S2. The emission data in other countries are obtained from
the IIASA emission inventory for 2015 (Zheng et al., 2019; Gao et al., 2020). The
biogenic emission is calculated online by the Model of Emissions of Gases and
Aerosols from Nature (MEGAN) v2.04 (Guenther et al., 2006). The dust emission
is calculated online by the Goddard Chemistry Aerosol Radiation and Transport
(GOCART) model coupled with the MOSAIC aerosol schemes. (Zhao et al., 2010;
2013)

To account for the physical processes of aerosol-radiation-cloud feedback
on meteorological factors and $PM_{2.5}$, the four-dimensional data assimilation
(FDDA) is not utilized in our simulations. Aerosol optical depth, single scattering
albedo, and asymmetry factors are calculated based on the Lorenz-Mie theory as

a function of wavelength and three-dimensional location (Fast et al., 2006). Then,
the aerosol optical properties are transferred to the RRTMG radiation scheme to
calculate the impact of aerosol on the radiation balance (Iacono et al., 2008). As
for the ACI, activated aerosols are calculated by the Abdul-Razzak and Ghan
scheme (Abdul-Razzak & Ghan, 2002) and are then coupled with the Morrison
two-moment cloud microphysics scheme (Morrison et al., 2009). The prognostic
cloud water content calculated by the Morrison scheme is input into the RRTMG
scheme for the radiative transfer calculation. It should be noted that the prognostic
aerosol does not influence cumulus clouds and ice nucleation in the model. The
prognostic aerosol can only be activated as CCN. It does not directly contribute
to ice nucleation, which is only influenced by air temperature and supersaturation
(Kanji et al., 2017). Furthermore, CCN would influence grid-scale clouds.
However, limited by the horizontal resolution of 27 km × 27 km, cumulus clouds
could not be resolved in this grid.

## 2.2 Experimental design

194        As described in the introduction, the purpose of this study is to quantify the

contributions of ARI and ACI to $PM_{2.5}$ concentrations under different emission
scenarios. The simulation periods are January and July, 2013 and 2021,
representing winter and summer, respectively.

198        As shown in Table 1, the enhanced $PM_{2.5}$ concentration induced by ARI and

ACI could be obtained via comparing the simulation results with ARI or ACI

turned on or off. By setting the 'aer_ra_feedback' to 0 in the model, the ARI could be turned off, which means that the interaction between aerosol and radiation is prevented. The ACI could be turned off through prescribing the CDNC of 25 cm$^{-3}$ in the microphysical scheme, which represents average level in the pristine air (Bennartz et al., 2007). For example, the 13M13E_B, 13M13E_NR and 13M13E_NRC shown in Table 1 represent the cases with ARI and ACI effects, without ARI effect, and without ARI and ACI effects in 2013, respectively. The ARI-induced $PM_{2.5}$ enhancement could be acquired by comparing the results of 13M13E_B and 13M13E_NR; the ACI-induced $PM_{2.5}$ enhancement could be obtained by comparing the results of 13M13E_NR and 13M13E_NRC.

**Table 1. Case definition under different meteorological backgrounds and anthropogenic emissions with ARI or ACI turned on or off.**

| Case | Meteorology | Emission | ARI | ACI |
|------|-------------|----------|-----|-----|
| 13M13E_B | Jan & Jul, 2013 | Jan & Jul, 2013 | on | on |
| 13M13E _NR | Jan & Jul, 2013 | Jan & Jul, 2013 | off | on |
| 13M13E _NRC | Jan & Jul, 2013 | Jan & Jul, 2013 | off | off |
| 21M13E_B | Jan & Jul, 2021 | Jan & Jul, 2013 | on | on |
| 21M13E_NR | Jan & Jul, 2021 | Jan & Jul, 2013 | off | on |
| 21M13E_NRC | Jan & Jul, 2021 | Jan & Jul, 2013 | off | off |
| 21M21E_B | Jan & Jul, 2021 | Jan & Jul, 2021 | on | on |
| 21M21E_NR | Jan & Jul, 2021 | Jan & Jul, 2021 | off | on |

| 21M21E_NRC | Jan & Jul, 2021 | Jan & Jul, 2021 | off | off |

213

In order to obtain the changes of the ARI- and ACI-induced $PM_{2.5}$ enhancements from 2013 to 2021 caused by the variation of meteorological background and by the reduction of anthropogenic emission, the control experiments (21M13E; three experiments: with ARI and ACI turned on, with ARI turned off and ACI turned on, and with ARI and ACI turned off) are designed with the meteorological background in 2021 and the anthropogenic emission in 2013. In the following, the 13M13E, 21M13E and 21M21E represent the cases with meteorological background and anthropogenic emission in 2013, meteorological background in 2021 and anthropogenic emission in 2013, and meteorological background and anthropogenic emission in 2021, respectively. Taking the ARI for example, the change of the ARI-induced $PM_{2.5}$ enhancement from the variation of meteorological background is obtained by subtracting the ARI-induced $PM_{2.5}$ enhancement in the 13M13E from that in the 21M13E [Eq. (1)]; the change in the ARI-induced $PM_{2.5}$ enhancement from the reduction of anthropogenic emission is obtained by subtracting the ARI-induced $PM_{2.5}$ enhancement in the 21M13E from that in the 21M21E [Eq. (2)]. The calculations for the ACI-induced $PM_{2.5}$ enhancement are similar, as shown in Eqs. (3) and (4).


$$ARI_{met} = (21M13E\_B - 21M13E\_NR) - (13M13E\_B - 13M13E\_NR), \qquad (1)$$

$$ARI_{emi} = (21M21E\_B - 21M21E\_NR) - (21M13E\_B -$$

$$21M13E\_NR), \tag{2}$$

$$ACI_{met} = (21M13E\_NR - 21M13E\_NRC) - (13M13E\_NR -$$

$$13M13E\_NRC), \tag{3}$$

$$ACI_{emi} = (21M21E\_NR - 21M21E\_NRC) - (21M13E_{NR} -$$

$$21M13E_{NRC}), \tag{4}$$

where the $ARI_{met}$ ($ACI_{met}$) and $ARI_{emi}$ ($ACI_{emi}$) represent the changes of the
enhanced $PM_{2.5}$ concentration induced by the ARI (ACI) from 2013 to 2021
caused by the variation of meteorological background and reduction of
anthropogenic emission, respectively.

## 2.3 Model evaluation

To determine the accuracy and reliability of simulation results, the
13M13E_B and 21M21E_B simulations (Table 1) are verified by using the
observations. The variables checked in the evaluation contain the concentration
and components of surface $PM_{2.5}$ and the meteorological factors, including air
temperature (T2) and water vapor mixing ratio (Q2) at 2 m, wind speed (WS10)
and wind direction (WD10) at 10 m, as well as cloud fraction (CF) and liquid
water path (LWP).
Simulated temperature, wind, and water vapor are compared with the
observations from the National Climate Data Center (NCDC,
http://www.ncdc.noaa.gov/). The evaluation shows that the absolute errors for T2,
WS10 and Q2 are respectively less than 1°C, 1 m s$^{-1}$ and 0.1 g kg$^{-1}$ (Table S3),
and those for WD10 are near or less than 10°. For the simulation utilizing the
FDDA, the benchmarks of biases proposed by Emery et al. (2001) are 0.7°C, 0.6
m s$^{-1}$, 1.0 g kg$^{-1}$ and 20° for the T2, WS10, Q2 and WD10, respectively. The
biases of the T2 and WS10 in our simulations have exceeded the benchmarks,
while they are still similar to or smaller than in most previous WRF-Chem
applications without FDDA over East Asia (Zhang et al., 2015; Zhao et al., 2017).
Simulated CF and LWP are compared with the data from the Moderate-
resolution Imaging Spectroradiometer (MODIS) aboard the Terra satellite
(http://ladsweb.nascom.nasa.gov/data/search.html). Overall, the CF and LWP
simulations are in good agreement with the observations (Figs. S1 and S2). The
high values of observed CF and LWP primarily appear in the south of China in
January 2013 and 2021, and high value of CF also occurs in the NCP region. The
high values of CF and LWP in the south of China could be reproduced in the
simulation, while the CF in NCP region is slightly underestimated, which could
be owing to imperfect cloud parameterization scheme in the model or
uncertainties in the retrieval of MODIS datasets. In July 2013 and 2021, part of
high value area of observed LWP and most high value area of observed CF appear
in the southwestern China and the east coast of China, which also could be
captured by the simulation. In addition, high LWP also appears in Gansu and
Sichuan Provinces in July 2013 and in the YRD and Sichuan-Chongqing in July
2021, which are both well reproduced. The distributions of low values of
observed CF and LWP in January and July of 2013 and 2021 are also well
simulated.
The simulation of surface $PM_{2.5}$ concentration is compared with the data
from the China National Environmental Monitoring Center
(*https://quotsoft.net/air/*). The evaluation shows that both the regional average
value and spatial distribution of simulated $PM_{2.5}$ concentration are in good
agreement with the observational data. As shown in Fig. S3, the biases of regional
average $PM_{2.5}$ concentration in January and July of 2013 and 2021 are below 3
$\mu g\ m^{-3}$ in eastern China. In this study, the eastern China includes most of Chinese
provinces except Xinjiang, Xizang, Ningxia, Qinghai, Gansu, Inner-Mongolia
and Heilongjiang Provinces, which contains most polluted regions in China. In
addition, the distributions of high simulated $PM_{2.5}$ concentration are also
consistent with the observations, such as the NCP region, the YRD region, and
the Sichuan-Chongqing area.
The simulated $PM_{2.5}$ components are also reasonable compared with the
observation data. Given that the $PM_{2.5}$ components data in 2013 are very rare, we
sourced three sets of data in January 2013, respectively in Beijing (Mattias et al.,
2017), Handan (Zhang et al., 2015), and Shanghai (Li et al., 2015). The results
show that the simulated $PM_{2.5}$ components are reproduced well generally.
Specifically, the simulated $PM_{2.5}$ components are larger than half of observational
$PM_{2.5}$ components and less than the double observational $PM_{2.5}$ components (Fig.
S4). Observed $PM_{2.5}$ components data in 2021 are from a data sharing platform
for the NCP region and its surrounding areas (Wang et al., 2019). Fig. S5 shows
the ratios of observation to simulation of ammonium, sulfate, BC and organic
carbon (OC) in January and July 2021. The results exhibit that almost all the ratios
of $PM_{2.5}$ components are located between 0.5 and 2.0, while some ratios of sulfate
in January, part of OC in January, and BC in January and July are beyond this
range. But these discrepancies will not cause obvious uncertainties in this
research. Specifically, considering BC low hygroscopicity, BC overestimations
in January and July 2021 probably bring low uncertainties in ACI-induced $PM_{2.5}$
enhancement. To test the impact of simulated BC overestimation in January 2021
on ARI-induced $PM_{2.5}$ enhancement, we utilize another set of particulate matter
(PM) source profiles (Liu et al., 2018) and conduct the simulations for January
2021. The results indicate that the ratios of simulated BC concentration to
observational BC concentration are within 2.0. The ARI-induced $PM_{2.5}$
enhancement is 1.33 $\mu g\ m^{-3}$, which shows a negligible difference from the result
(1.37 $\mu g\ m^{-3}$) obtained using original PM source profiles (Fig. S6). In view of the
results in January 2021, the BC overestimation in July 2021 also probably brings
low uncertainties in ARI-induced $PM_{2.5}$ enhancement. However, the reduction in
simulated BC concentration in January 2021 does not necessarily mean that this
set of PM source profiles is better than the original PM source profiles, because
this might be an accidental result caused by other uncertainties. For example, the
current model underestimates the wet deposition of BC due to neglecting the
increase in BC hygroscopicity brought about by BC aging. If this process is

considered in the model, simulated BC concentrations might be better reproduced using original PM source profiles. Therefore, in this study, we still use the original results for our analysis. The model also underestimates the sulfate concentration and overestimates the part of OC concentration in January 2021. We think that neither of these discrepancies will cause significant uncertainties in ARI- and ACI-induced $PM_{2.5}$ enhancement. Specifically, the majority of aerosol is scattering aerosol and the $PM_{2.5}$ concentration in January 2021 is reproduced well. Therefore, we think that the impact of the sulfate underestimation on the ARI-induced $PM_{2.5}$ enhancement would be largely offset by the overestimation of other scattering aerosol components, such as OC. In addition, the OC overestimation should not bring significant uncertainty to ACI-induced $PM_{2.5}$ enhancement either, because of the relatively lower hygroscopicity of OC compared to secondary inorganic aerosol. The underestimation of sulfate simulation in January 2021 also minimally affects ACI-induced $PM_{2.5}$ enhancement because the sulfate underestimation mainly occurs in the North China Plain, where cloud cover is low. In contrast, in southern cities such as Mianyang city in Sichuang province where there is plenty of cloud cover, the sulfate simulation was 4.19 μg m$^{-3}$ in January 2021, which is very close to the observed value of 4.25 μg m$^{-3}$ (Lin et al., 2022).

In summary, the performances of WRF-Chem model on the simulations of air quality and meteorological factors over China are fairly good, and the differences between simulations and observations are reasonable and acceptable.

## 3. Results and discussion

## 3.1 The impacts of ARI and ACI feedbacks on the meteorological factors and PM$_{2.5}$ concentrations in 2013

We comprehensively discuss the effects of ARI and ACI on the regional meteorological factors and PM$_{2.5}$ concentrations in January and July 2013. Fig. 1 shows the impacts of ARI and ACI feedbacks on the SWDOWN, PBLH, T2, RH and PM$_{2.5}$ concentration in January and July 2013. For the ARI, the SWDOWN decreases by 18.37 and 7.71 W m$^{-2}$ in January and July 2013 in eastern China, respectively. Since the incoming solar radiation reaching the ground is reduced by PM, the T2 and PBLH in eastern China further decrease by 0.30 and 0.03℃, and 28.34 and 8.75 m in January and July 2013, respectively. Meanwhile, the RH increases by 0.46% and 0.08% due to the water vapor accumulation in the suppressed planetary boundary layer (Liu et al., 2018). Ultimately, the PM$_{2.5}$ concentration increases by 5.59 and 0.13 μg m$^{-3}$ in eastern China (Fig. 1d). For the ACI, affected by the cloud modified by the aerosol, the SWDOWN, T2 and PBLH decrease by 7.54 and 14.03 W m$^{-2}$, 0.18 and 0.17 ℃, and 10.89 and 24.31 m, and the RH increases by 0.34% and 0.37% in January and July 2013 in eastern China, respectively. As a result, the PM$_{2.5}$ concentration increases by 3.96 and 2.20 μg m$^{-3}$ in eastern China. Fig. 2 shows that the regional averaged values and spatial distributions of PM$_{2.5}$ enhancements induced by ARI and ACI in 2013 are in line with the results of previous studies (Zhao et al., 2017; Zhang et al., 2018).

358   Overall, the enhanced PM$_{2.5}$ concentration induced by ARI is greater than

359 that induced by ACI in January 2013, which is due to the relatively low LWP in

360 the high PM$_{2.5}$ concentration area. But it shows the opposite situation in July 2013,

361 owing to the plentiful cloud in warm July (Zhang et al., 2018).

362

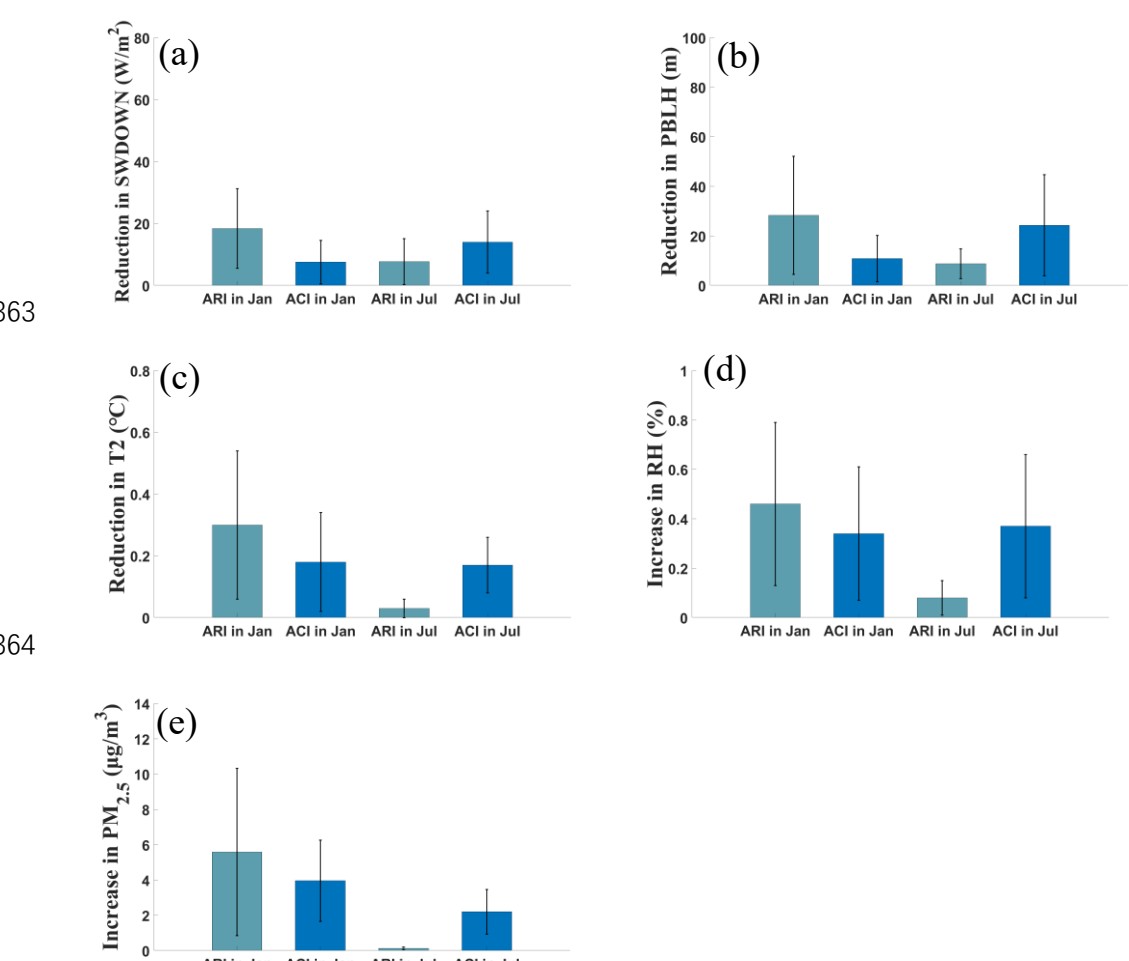

Fig. 1. The regional averaged reductions of (a) downward shortwave radiation at the surface (SWDOWN), (b) planetary boundary layer height (PBLH), (c) 2-m air temperature (T2), and increments of (d) relative humidity (RH) and (e) fine particulate matter (PM$_{2.5}$) concentration induced by the aerosol-radiative interaction (ARI) and aerosol-cloud interaction (ACI) in January and July 2013

in eastern China, the error bars represent the standard deviations for different
meteorological factors and $PM_{2.5}$ concentration induced by ARI and ACI in
January and July 2013 in eastern China.

## 3.2 The shift of the $PM_{2.5}$ enhancements induced by ARI and ACI

As discussed in section 3.1, the enhanced $PM_{2.5}$ concentrations induced by
ARI and ACI exhibit obvious spatial and seasonal variations in 2013. However,
due to the variations of meteorological background and the reduction of
anthropogenic emission from 2013 to 2021, their joint and individual impacts on
the ARI- and ACI-induced $PM_{2.5}$ enhancements are still unclear. Fig. 2 shows the
ARI- and ACI-induced $PM_{2.5}$ enhancements in the experiments of 13M13E,
21M13E and 21M21E in January and July.

As shown in Fig. 2, from 2013 to 2021, the $PM_{2.5}$ concentration
enhancement induced by the ARI in January decreases by 75% (from 5.59 to 1.37
$\mu g\ m^{-3}$). Zhang et al. (2022) also found that the ARI effect over China weakens
during 2013–2017, and the ratio of $PM_{2.5}$ enhancement to the ambient $PM_{2.5}$
concentration decreases from 5.40% to 3.30%. The decline of the $PM_{2.5}$
enhancement ratio (2.10%) is lower than that in this study (3.26%) due to the
continuous emission reduction after 2017. On the other hand, the ACI-induced
$PM_{2.5}$ enhancement decreases by 51%, from 3.96 to 1.93 $\mu g\ m^{-3}$. With lower
percentage decrease in the $PM_{2.5}$ enhancement, the ACI-induced $PM_{2.5}$
enhancement exceeds the ARI-induced $PM_{2.5}$ enhancement in January 2021. In

July, both the ARI- and ACI-induced $PM_{2.5}$ enhancements show decreasing trends,
the percentage decreases of the ARI-induced (31%) and ACI-induced (34%)
$PM_{2.5}$ enhancements are very close.
The contributions of the meteorological background variation and
anthropogenic emission reduction to the changes of the ARI- and ACI-induced
$PM_{2.5}$ enhancements are different. Due to the meteorological background change
from 2013 to 2021, the ARI- and ACI-induced $PM_{2.5}$ enhancements show
different characteristics in January and July. It can be seen that, the ARI-induced
$PM_{2.5}$ enhancement decreases from 5.59 to 3.15 $\mu g\ m^{-3}$ with the variation of
meteorological background in January, while it increases from 0.13 to 0.27 $\mu g\ m$
$^{-3}$ in July. The primary reason for the difference is that the ambient $PM_{2.5}$
concentration decreases in January but increases in July caused by different
meteorological backgrounds. The ACI-induced $PM_{2.5}$ enhancement changes
slightly from 3.96 to 3.40 $\mu g\ m^{-3}$ in January due to the variation of meteorological
background. However, it increases from 2.20 to 3.31 $\mu g\ m^{-3}$ in July, because of
a large aerosol-induced LWP increase in July 2021.
Considering the reduction of anthropogenic emission, the ARI- and ACI-
induced $PM_{2.5}$ enhancements both show declining trends (middle and right
columns in Fig. 2). The ARI-induced $PM_{2.5}$ enhancement decreases by 56.51% in
January, from 3.15 to 1.37 $\mu g\ m^{-3}$ . The ACI-induced $PM_{2.5}$ enhancement
decreases by 43.24%, from 3.40 to 1.93 $\mu g\ m^{-3}$. The percentage decrease of the
ACI-induced $PM_{2.5}$ enhancement is lower than that of the ARI-induced in January,

which also occurs in July, when the ARI-induced $PM_{2.5}$ enhancement decreases by 66.67% (from 0.27 to 0.09 μg m$^{-3}$) and ACI-induced $PM_{2.5}$ enhancement decreases by 56.50% (from 3.31 to 1.44 μg m$^{-3}$).

In summary, both the variation of meteorological background and the reduction of anthropogenic emission play important roles in changing the ARI- and ACI-induced $PM_{2.5}$ enhancements. However, the decreases of ARI- and ACI-induced $PM_{2.5}$ enhancements from 2013 to 2021 are primarily attributed to the reduction of anthropogenic emission. In addition, the percentage decrease of the ACI-induced $PM_{2.5}$ enhancement is lower than that induced by the ARI in both January and July. Therefore, the ACI-induced $PM_{2.5}$ enhancement has become increasingly important in both January and July from 2013 to 2021.

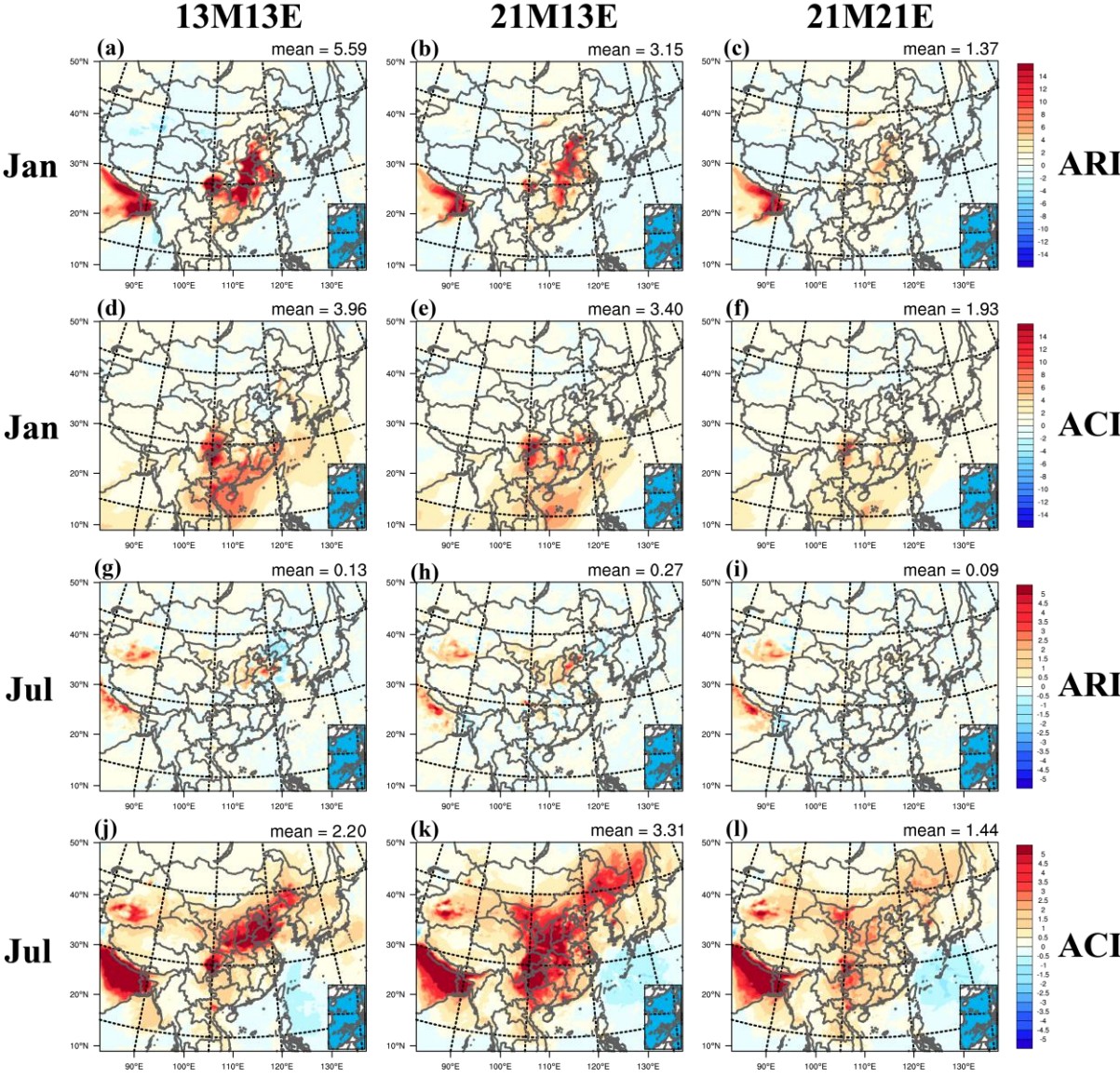

Fig. 2. The distributions of enhanced $PM_{2.5}$ concentrations (unit: μg m$^{-3}$) induced by the ARI (first and third rows) and the ACI (second and fourth rows) in January (first and second rows) and July (third and fourth rows) in the experiments of 13M13E (left column), 21M13E (middle column) and 21M21E (right column).

## 3.3 The changes in the enhanced $PM_{2.5}$ components induced by the ARI and the ACI

In terms of the anthropogenic emission reduction, the percentage decrease

of the ACI-induced PM$_{2.5}$ enhancement is lower than that induced by the ARI in
both January and July. We find that the difference is primarily from the different
percentage decreases of the secondary PM$_{2.5}$ component enhancements induced
by ARI and ACI.
Fig. 3 shows the percentage decreases of ARI- and ACI-induced PM$_{2.5}$
component enhancements caused by the anthropogenic emission reduction in
January and July. It can be seen that the difference between the percentage
decreases of the ARI- and ACI-induced enhancements of sulfate, nitrate,
ammonium and OC is larger than those of BC and other inorganic aerosol (OIN).
OIN refers to inorganic compositions other than sulfate, nitrate, ammonium, and
BC. These compositions include sea salt and mineral elements. Specifically, the
difference between the percentage decreases for sulfate, nitrate, ammonium and
OC enhancements are 34.66%, 40.20%, 13.80% and 25.65% respectively, and the
values for OIN and BC are 8.67% and 6.67%. This result indicates that the lower
decrease in the ACI-induced PM$_{2.5}$ concentration enhancement is mainly due to
the small decrease in the ACI-induced enhancements of secondary PM$_{2.5}$
components. The main causes will be illustrated in section 3.4.

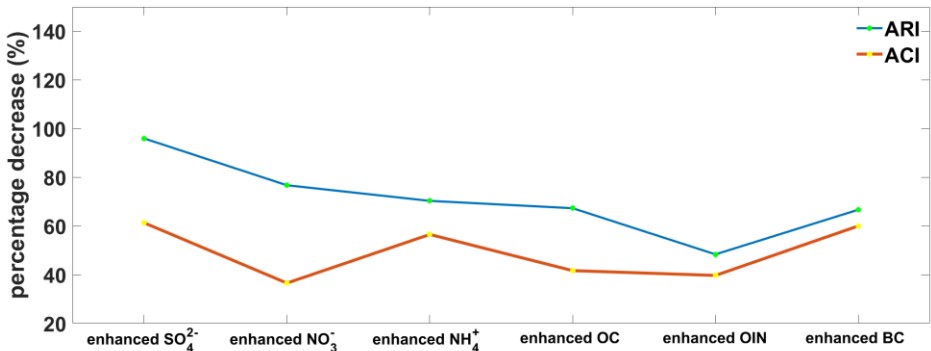

Fig. 3. Percentage decreases (21M13E−21M21E)/21M13E) of the spatial and temporal average ARI- and ACI-induced $PM_{2.5}$ component enhancements in eastern China in January and July caused by the anthropogenic emission reduction from 2013 to 2021.

## 3.4 Causes for the increased importance of ACI

### 3.4.1 Explanation from the perspective of meteorological changes

As discussed in previous studies, the decrease of PBLH and T2 and the increase of RH are tightly related to the ARI- and ACI-induced $PM_{2.5}$ enhancements (Donahue et al., 2012; Ding et al., 2016; Moch et al., 2022; Liu et al., 2018). From the perspective of the ARI- and ACI-induced changes in meteorological factors, we investigate the primary reasons for the increasing importance of the ACI-induced $PM_{2.5}$ enhancement under the reduction of anthropogenic emission.

Fig. 4 shows the percentage decreases of ARI- and ACI-induced decrease of SWDOWN, PBLH and T2 and increase of RH due to the reduction of anthropogenic emission from 2013 to 2021. In January, in order to illustrate the

reasons of the lower percentage decrease in the ACI-induced $PM_{2.5}$ enhancement
clearly, we take the highly polluted NCP region as an example. As shown in Fig.
4c, the percentage decreases of the ACI-induced decline of SWDOWN (19%),
PBLH (27%) and T2 (20%) and the increase of RH (24%) are lower than those
of the ARI-induced decline of SWDOWN (29%), PBLH (39%) and T2 (32%) and
the increase of RH (36%). The phenomenon in July is similar with that in January
(Figs. 4a and b). To our knowledge, the PBLH and T2 are determined by the
incoming solar radiation at the surface, and they can strongly influence the RH.
So the lower percentage decrease in the ACI-induced reductions of PBLH and T2
and increase of RH could be explained by the lower percentage decrease in the
ACI-induced SWDOWN reduction.
We believe that the relatively lower decrease in the ACI-induced SWDOWN
reduction is inevitable under high ambient $PM_{2.5}$ concentration. As shown in Fig.
S8b, the SWDOWN reduction induced by the ARI shows a linear relationship
with the decline of ambient $PM_{2.5}$ concentration, which is similar with Zhou et al.
(2018). In contrast, the decrease in the SWDOWN reduction induced by the ACI
is lower than that by the ARI due to the ambient $PM_{2.5}$ decrease in the high $PM_{2.5}$-
polluted regime. The reason is that the decrease in ambient $PM_{2.5}$ concentration
directly weakens the ARI-induced SWDOWN reduction, but it has only a minor
impact on the ACI-induced SWDOWN reduction because the change in LWP and
cloud effective radius (Re) induced by ACI is not sensitive to $PM_{2.5}$ reduction in
the $PM_{2.5}$-polluted regime. In our simulations, the influence of ACI-induced Re

change is relatively smaller than that of ACI-induced LWP change with a large

decrease in $PM_{2.5}$ concentration (Fig. S7). Therefore, we are only concerned with

change in ACI-induced LWP with a reduction in $PM_{2.5}$. As shown in Fig. S8a,

when the ambient $PM_{2.5}$ concentration exceeded 15 μg m$^{-3}$, the decrease in ACI-

induced LWP increase is relatively low with a $PM_{2.5}$ reduction from 120 to 15 μg

m$^{-3}$, indicating that aerosols are not a key limiting factor to cloud formation in

this range. Note that when the ambient $PM_{2.5}$ concentration decreases to 15 μg

m$^{-3}$, the weakening of SWDOWN reduction induced by the ACI might be larger

than that by the ARI. This is because decrease in ACI-induced LWP increase is

relatively fast, with a $PM_{2.5}$ reduction from 15 to 0 μg m$^{-3}$. Previous studies have

demonstrated that the decrease in ACI-induced LWP increase is relatively fast or

slow with the ambient $PM_{2.5}$ reduction in the $PM_{2.5}$-clean or polluted condition,

respectively (Myhre et al., 2007; Savane et al., 2015). The regional and temporal

average $PM_{2.5}$ concentration in eastern China in January and July simulated using

background meteorology in 2021 and emissions in 2013 is 63 and 25 μg m$^{-3}$,

which is much higher than 15 μg m$^{-3}$. Therefore, the decrease in ACI-induced

SWDOWN reduction in both months is weak.

Especially, the lower PBLH caused by ARI and ACI will enhance the

accumulation of all the $PM_{2.5}$ components, but higher RH and lower T2 induced

by the ARI and ACI could promote the production of extra secondary $PM_{2.5}$

components through strengthening aqueous and heterogeneous reactions and

causing gas precursors to condense into particle matter (Donahue et al., 2012; Liu

et al., 2018). Therefore, lower percentage decrease in the T2 reduction and RH
increase induced by the ACI is more likely to weaken the decrease in the
enhancements of secondary $PM_{2.5}$ components. This well explains the lower
percentage decreases in the enhancements of secondary $PM_{2.5}$ components
induced by the ACI than those by the ARI as shown in Fig. 3.

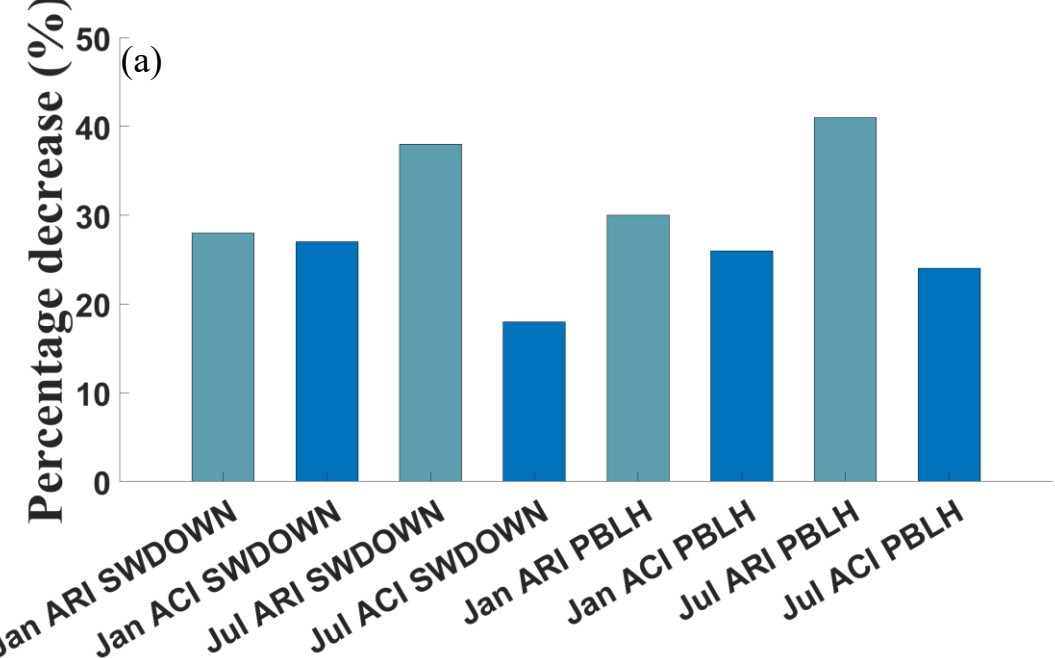



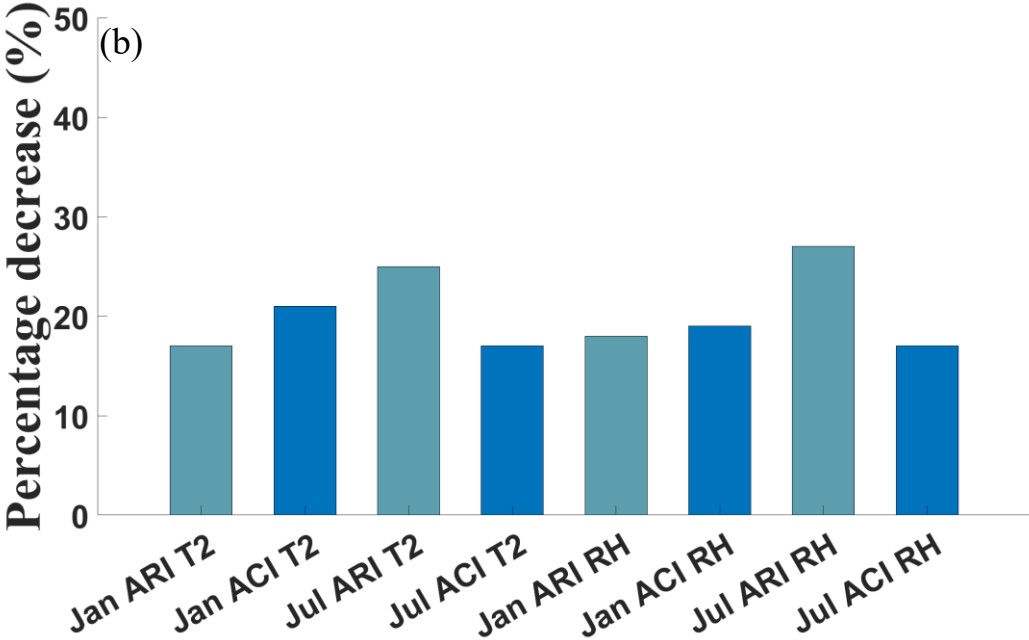


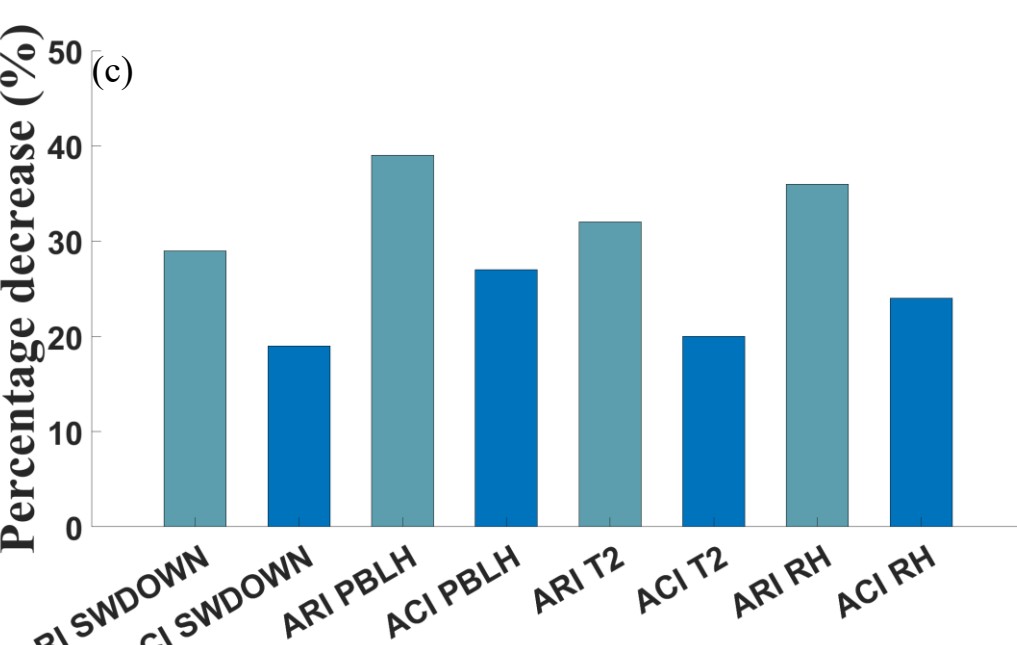


Fig. 4. The percentage decreases of the regional averages of (a) the decrease of

SWDOWN and PBLH, and (b) the T2 reduction and RH increase induced by ARI

and ACI in eastern China caused by the anthropogenic emission reduction in

January and July from 2013 to 2021. (c) is the same as (a) and (b), but in the NCP

region in January.

531

**3.4.2 Explanation from the perspective of PM$_{2.5}$ concentration distribution changes**

Ambient PM$_{2.5}$ concentration is the fundamental factor to trigger the ARI and the ACI. In order to further explore the reasons for the increasing importance of enhanced PM$_{2.5}$ concentration induced by ACI, we discuss the characteristics of enhanced PM$_{2.5}$ concentration induced by ARI and ACI under different PM$_{2.5}$ pollution levels. Given that this study mainly focuses on the change in ARI- and ACI-induced PM$_{2.5}$ enhancement in the PM$_{2.5}$-polluted regime. we only discuss these changes within the PM$_{2.5}$ concentration range of 15–180 μg m$^{-3}$

The PM$_{2.5}$ concentration is divided into 11 levels from 15 to 180 μg m$^{-3}$. As shown in Fig. 5a, in the heavily PM$_{2.5}$-polluted regime (135–180 μg m$^{-3}$), the decrease in SWDOWN induced by ARI is much larger than that induced by ACI (Fig. S9a). Then, the decrease in PBLH and T2 and the increase in RH induced by ARI are also larger than those induced by ACI (Fig. S9b–d). Thus, the enhanced PM$_{2.5}$ induced by the ARI is much larger than that by the ACI (Fig. 5a). However, when the PM$_{2.5}$ concentration decrease to the range of 15–45 μg m$^{-3}$, the decrease in SWDOWN, PBLH, and T2 and the increase in RH induced by ACI significantly exceed those induced by ARI. Thus, the ACI-induced PM$_{2.5}$ enhancement significantly exceeds the ARI-induced PM$_{2.5}$ enhancement and becomes more important. This indicates the fast decrease in the ARI-induced PM$_{2.5}$ enhancement and the increasing contribution of the ACI-induced PM$_{2.5}$

enhancement with the decrease in the PM$_{2.5}$ concentration. In summary, the
percentage decrease in the PM$_{2.5}$ enhancement induced by ACI is weaker than
that induced by ARI with the decrease of PM$_{2.5}$ concentration because of the
lower percentage decrease in the ACI-induced SWDOWN, which causes the
lower percentage decrease in the ACI-induced PBLH and T2 reduction and the
RH increase. Furthermore, as shown in Fig. S8a, the low percentage decrease in
the ACI-induced SWDOWN reduction is due to a low decrease in the ACI-
induced LWP in the PM$_{2.5}$-polluted regime. Considering the decrease in the
ambient PM$_{2.5}$ concentration due to the anthropogenic emission reduction from
2013 to 2021 (Fig. 5b), the ACI-induced PM$_{2.5}$ enhancement certainly contributes
more to the total PM$_{2.5}$ concentration in 2021.

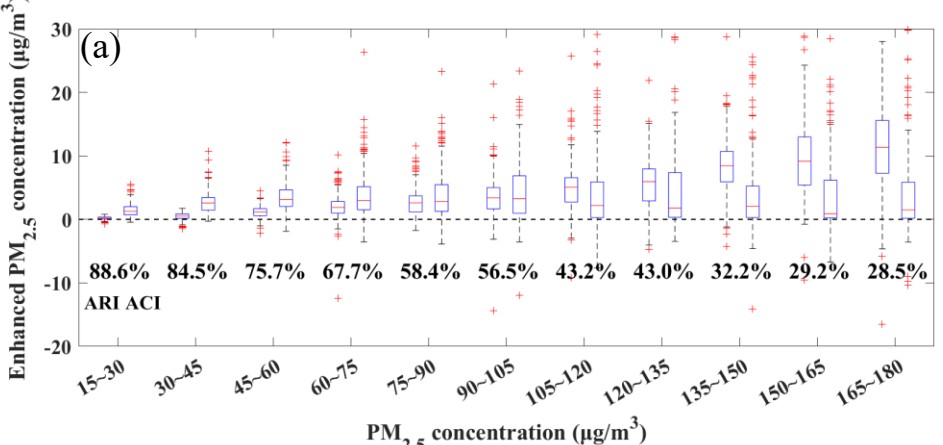


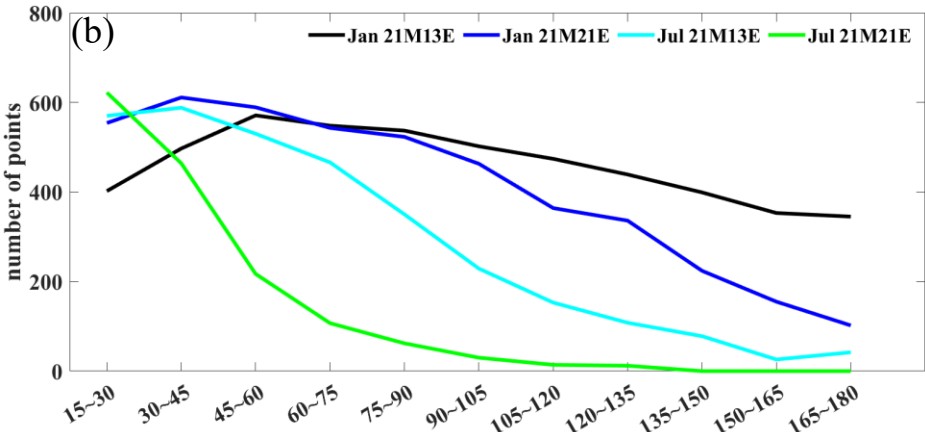

Fig. 5. (a) The enhanced PM$_{2.5}$ concentrations induced by ARI and ACI at different ambient PM$_{2.5}$ levels. These data are from the simulations for January and July in the experiments of 21M13E and 21M21E. The percentage represents the ratio of the ACI-induced PM$_{2.5}$ enhancement to the sum of ARI- and ACI-induced PM$_{2.5}$ enhancements. (b) The distributions of ambient PM$_{2.5}$ levels in January and July in the experiments of 21M13E and 21M21E.

## 4. Conclusions

Under the background of sharped anthropogenic emission reduction, this study investigates changes of the ARI- and ACI-induced PM$_{2.5}$ enhancements for 2013–2021, and explores the causes for these changes from the perspectives of meteorological factors and PM$_{2.5}$ concentration distribution.

The results show that the enhanced PM$_{2.5}$ induced by the ARI (5.59 μg m$^{-3}$) is greater than that by the ACI (3.96 μg m$^{-3}$) in January 2013. However, the ARI- and ACI-induced PM$_{2.5}$ enhancements decrease from 5.59 and 3.96 μg m$^{-3}$ to 1.37 and 1.93 μg m$^{-3}$ in January and decrease by 75% and 51% for 2013–2021. The

smaller decrease ratio (51%) for ACI-induced $PM_{2.5}$ enhancements implies that ACI becomes more important for enhancing $PM_{2.5}$ concentrations in January 2021. Furthermore, we separated the contributions of meteorological background variation and anthropogenic emission reduction. Compared with the meteorological background variation, anthropogenic emission reduction plays a more important role in causing the decrease of ARI- and ACI-induced $PM_{2.5}$ enhancements. Owing to only emission reduction, the enhanced $PM_{2.5}$ concentrations induced by the ARI and ACI decrease by 56% and 43% in January and 66% and 56% in July, respectively. The ACI-induced $PM_{2.5}$ enhancement becomes increasingly important in both January and July for 2013–2021. More specifically, the lower percentage decrease in the ACI-induced $PM_{2.5}$ enhancement is dominated by the lower decrease in the enhancements of secondary $PM_{2.5}$ components.

The lower percentage decrease in the enhanced $PM_{2.5}$ induced by the ACI is due to the lower percentage decrease in the ACI-induced SWDOWN reduction, which is because of the lower decrease in the LWP and increase in the Re caused by the ambient $PM_{2.5}$ decrease in the high $PM_{2.5}$-polluted regime (Fig. 6). At the same time, the lower percentage decreases in the T2 reduction and RH increase induced by the ACI further lead to the lower percentage decrease in the enhancements of the ACI-induced secondary $PM_{2.5}$ components (Fig. 6). Notably, due to relative lower percentage decrease in the ACI-induced SWDOWN reduction in the high $PM_{2.5}$-polluted regime, the increasing importance of ACI-

induced PM$_{2.5}$ enhancement is a matter of course with the ambient PM$_{2.5}$ decrease.

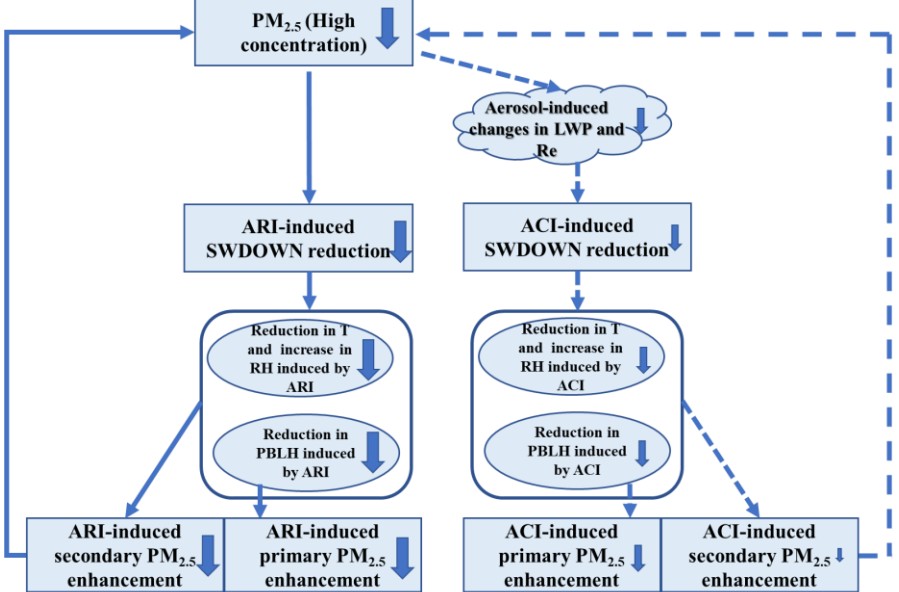


Fig. 6. Schematic diagram for the decrease of ARI- and ACI-induced primary and
secondary PM$_{2.5}$ enhancement due to reduction in ambient PM$_{2.5}$ concentration.
Solid arrows represent these processes are strongly weakened; dotted arrows
represent these processes are slightly weakened.

613   This study has important implication for the PM$_{2.5}$ control. As we know,

ARI- and ACI-induced PM$_{2.5}$ enhancements have a non-negligible contribution
to the deterioration of PM$_{2.5}$ air quality. Previous research has investigated the
impact of anthropogenic emission reduction on the ARI-induced PM$_{2.5}$
enhancement (Zhou et al., 2019). But compared with PM$_{2.5}$ enhancement induced
by ARI, that induced by ACI is more complicated and harder to be alleviated. Our
findings have further revealed that the ACI-induced PM$_{2.5}$ enhancement is getting
more important relative to that induced by ARI. This is especially true in cloud-

prone areas like Sichuan-Chongqing area, which have witnessed rather weak decreases of ACI-induced $PM_{2.5}$ concentration in the past decade due to weak decreases of aerosol-induced LWP under the condition of high ambient $PM_{2.5}$ level (Fig. 2). The ACI-induced $PM_{2.5}$ enhancement needs to be considered more seriously in the formulation of control polices to meet national $PM_{2.5}$ air quality standard, especially in cloud-prone areas with high ambient $PM_{2.5}$ concentration. To control ACI-induced $PM_{2.5}$ enhancement, first, a larger emission reduction is necessary in cloudy areas compared with less cloudy areas to bring about a noticeable decrease in ACI-induced LWP in response to $PM_{2.5}$ reduction. Second, secondary inorganic aerosol (SNA), which is an important component of total aerosol, has a large influence on the ACI-induced $PM_{2.5}$ enhancement because of its high hygroscopicity. This makes it easy for SNA to be activated as CCN and influence LWP. We think that it is crucial to make substantial decreases in the precursors of SNA, such as $SO_2$, $NO_x$ and $NH_3$ species. These decreases could substantially decrease SNA. A large decrease in SNA would enhance the ACI-induced LWP response to $PM_{2.5}$ reduction and cause a large decrease in ACI-induced $PM_{2.5}$ enhancement. In addition, relative to ARI-induced $PM_{2.5}$ enhancement, the lower decrease in ACI-induced $PM_{2.5}$ enhancement is mainly because of the small decrease in ACI-induced enhancements of secondary $PM_{2.5}$ components. A substantial decrease in SNA would make the decrease ratio of ACI-induced $PM_{2.5}$ enhancement approach the more rapid decrease ratio of ARI-induced $PM_{2.5}$ enhancement.

**Data and Code availability.**

The data and code used in this study are available upon request from Da Gao (dagao94@foxmail.com).

**Author Contribution**

D.G., B.Z. and S.W. designed the research; D.G., B.Z., J.S. and B.G. improved the WRF-Chem performance; D.G. and B.Z. further developed WRF-Chem and performed the simulations; X.W., S.L. and Z.D. provide the anthropogenic emissions; D.G. analyzed the data with the help from B.Z., S.W. and Y.W.; D.Y. and J.S. help D.G. to design some figures; S.W., Y.W., Y.Z. and Y.H. presented important suggestions for the analysis and writings; D.G. and B.Z. wrote the paper with inputs from all co-authors.

**Competing interests**

The author declares no competing interests.

**Acknowledgments**.

This research is supported by the National Key Research and Development Program of China (2022YFC3701000, Task 5), the National Natural Science Foundation of China (22188102), and the Tencent Foundation through the XPLORER PRIZE. We would like to thank Fenfen Zhang for providing the $PM_{2.5}$

components data for Handan city in January 2013.

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
