# Peer review of "China with the Anthropogenic Emission Reduction"

_EGUsphere, 2023_

## Author Response (AR1)

**A point-to-point response to the referee's comments**

Reply to anonymous referee #1

On behalf of my co-authors, we would like to express our great appreciation for your constructive comments and feedback on our manuscript entitled "Increased Importance of Aerosol-Cloud Interaction for Surface PM$_{2.5}$ Pollution Relative to Aerosol-Radiation Interaction in China with Anthropogenic Emission Reduction" (egusphere-2023-502). We have studied your comments carefully and have revised our manuscript accordingly. Replies to comments are in blue, and the line numbers in the response refer to the clean revised manuscript.

This study attempts to evaluate the individual contributions of aerosol-cloud interaction (ACI) and aerosol-radiation interaction (ARI) to the enhancement of PM$_{2.5}$ under the anthropogenic emission reduction scenario. The manuscript is well written and the topic is important at current stage. I recommend it for publication with some minor revisions, which are listed as below.

Response: Thank you for your careful evaluation of this manuscript. All your comments and suggestions are important to us. They have direct significance to our paper writing and research work.

Specific comments

From Figs. 1 and 2, the PM$_{2.5}$ increase caused by ARI in July appears quite minor compared to that caused by ACI. From the perspective of meeting the national standard of PM$_{2.5}$, changes in absolute values in PM$_{2.5}$ concentrations make more sense, particularly in July. As such, it seems not so fair to stress too much on the relative change of PM$_{2.5}$ enhancement in July when comparing ARI and ACI feedbacks, for example, in the Abstract (Lines 37-41) and other places in body text. Also in Abstract, it only January mentioned in Lines 34-37 but both January and July were mentioned later (Lines 37-41), making the statements not consistent. Is there a similar shift in July as stated in Lines 34-37 for January? If not, it might not be necessary to emphasize the results of July here.

Response: Thank you for your comment. In the paper, we discuss the results in July to make our conclusion more robust and reliable. The phenomenon of aerosol-cloud interactions (ACI) is becoming more important in terms of enhancing PM$_{2.5}$ concentrations both in January (the most PM$_{2.5}$-polluted month in a year) and July (the least PM$_{2.5}$-polluted month in a year). This indicates that it is not an accidental phenomenon and probably occurs in every month. In addition, the increasing

importance of the ACI-induced PM$_{2.5}$ enhancement does not contradict the original importance of ACI-induced PM$_{2.5}$ enhancement in July. Therefore, we think that the results in July still need to be discussed in the body text.

In the abstract, following reviewer's suggestion, we emphasize the results in January. Between January and July, the percentage decrease of the ACI-induced PM$_{2.5}$ enhancement is consistently weaker than the ARI-induced PM$_{2.5}$ enhancement owing to emission reduction. However, as the reviewer mentioned, the ARI-induced PM$_{2.5}$ enhancement is small in July, and thus it is not emphasized in the abstract. The revised abstract wording is as follows:

Abstract: Surface fine particulate matter (PM$_{2.5}$) pollution can be enhanced by feedback processes induced by aerosol-radiation interactions (ARI) and aerosol-cloud interactions (ACI). Many previous studies have reported enhanced PM$_{2.5}$ concentration induced by ARI and ACI for episodic events in China. However, few studies have examined the changes in the ARI- and ACI-induced PM$_{2.5}$ enhancements over a long period, though the anthropogenic emissions have changed substantially in the last decade. In this study, we quantify the ARI- and ACI-induced PM$_{2.5}$ changes for 2013–2021 under different meteorology and emission scenarios using the Weather Research and Forecasting model with Chemistry (WRF-Chem) and investigate the driving factors for the changes. Our results show that in January 2013, when China suffered from the worst PM$_{2.5}$ pollution, the PM$_{2.5}$ enhancement induced by ARI in eastern China (5.59 $\mu$g m$^{-3}$) is larger than that induced by ACI (3.96 $\mu$g m$^{-3}$). However, the ACI-induced PM$_{2.5}$ enhancement shows a significantly smaller decrease ratio (51%) than the ARI-induced enhancement (75%) for 2013–2021, making ACI more important for enhancing PM$_{2.5}$ concentrations in January 2021. Our analyses suggest that the anthropogenic emission reduction plays a key role in this shift. Owing to only anthropogenic emission reduction, the ACI-induced PM$_{2.5}$ enhancement decreases by 43% in January, lower than the decrease ratio of the ARI-induced enhancement (57%). The relative change in ARI- and ACI-induced PM$_{2.5}$ enhancement in July is similar to the pattern observed in January caused by anthropogenic emission reduction. The primary reason for this phenomenon is that the decrease of ambient PM$_{2.5}$ for 2013–2021 causes a disproportionately small decrease of liquid water path (LWP) and increase of cloud effective radius (Re) under the condition of high PM$_{2.5}$ concentration. Therefore, the surface solar radiation attenuation (and hence boundary layer height reduction) caused by ACI decreases slower than that caused by ARI. Moreover, the lower decrease ratio of the ACI-induced PM$_{2.5}$ enhancement is dominated by the lower decrease ratio of ACI-induced secondary PM$_{2.5}$ component enhancement, which is additionally caused by smaller decrease ratio of the air temperature reduction and relative humidity (RH) increase. Our findings indicate that, with the decrease of ambient PM$_{2.5}$, the ACI-induced PM$_{2.5}$ enhancement inevitably becomes more important. This needs to be considered in the formulation of control policies to meet the national PM$_{2.5}$ air quality standard.

Lines 545-547: Could the authors provide some details about the possible implications on emissions control policies over cloud-prone area given that ACI becomes more important in $PM_{2.5}$ enhancement relative to ARI with $PM_{2.5}$ emission reduction?

Response: Thank you for your comment. Compared with less cloudy areas, the ACI-induced $PM_{2.5}$ enhancement is important in cloudy areas. With the emission reduction, the ACI-induced $PM_{2.5}$ enhancement becomes more important relative to the ARI-induced $PM_{2.5}$ enhancement. To control ACI-induced $PM_{2.5}$ enhancement, first, a larger emission reduction is necessary in cloudy areas compared with less cloudy areas to bring about a noticeable decrease in ACI-induced LWP in response to $PM_{2.5}$ reduction. Second, secondary inorganic aerosol (SNA), which is an important component of total aerosol, has a large influence on the ACI-induced $PM_{2.5}$ enhancement because of its high hygroscopicity. This makes it easy for SNA to be activated as CCN and influence LWP. We think that it is crucial to make substantial decreases in the precursors of SNA, such as $SO_2$, $NO_x$ and $NH_3$ species. These decreases could substantially decrease SNA. A large decrease in SNA would enhance the ACI-induced LWP response to $PM_{2.5}$ reduction and cause a large decrease in ACI-induced $PM_{2.5}$ enhancement. In addition, relative to ARI-induced $PM_{2.5}$ enhancement, the lower decrease in ACI-induced $PM_{2.5}$ enhancement is mainly because of the small decrease in ACI-induced enhancements of secondary $PM_{2.5}$ components. A substantial decrease in SNA would make the decrease ratio of ACI-induced $PM_{2.5}$ enhancement approach the more rapid decrease ratio of ARI-induced $PM_{2.5}$ enhancement.

The implications for controlling ACI-induced $PM_{2.5}$ enhancement have been added in the revised manuscript. Please see lines 624-642.

Lines 624-642: The ACI-induced PM2.5 enhancement needs to be considered more seriously in the formulation of control polices to meet national PM2.5 air quality standard, especially in cloudy areas with high ambient $PM_{2.5}$ concentration. To control ACI-induced $PM_{2.5}$ enhancement, first, a larger emission reduction is necessary in cloudy areas compared with less cloudy areas to bring about a noticeable decrease in ACI-induced LWP in response to $PM_{2.5}$ reduction. Second, secondary inorganic aerosol (SNA), which is an important component of total aerosol, has a large influence on the ACI-induced $PM_{2.5}$ enhancement because of its high hygroscopicity. This makes it easy for SNA to be activated as CCN and influence LWP. We think that it is crucial to make substantial decreases in the precursors of SNA, such as $SO_2$, $NO_x$ and $NH_3$ species. These decreases could substantially decrease SNA. A large decrease in SNA would enhance the ACI-induced LWP response to $PM_{2.5}$ reduction and cause a large decrease in ACI-induced $PM_{2.5}$ enhancement. In addition, relative to ARI-induced $PM_{2.5}$ enhancement, the lower decrease in ACI-induced $PM_{2.5}$ enhancement is mainly because of the small decrease in ACI-induced enhancements of secondary $PM_{2.5}$ components. A substantial decrease in SNA would make the decrease ratio of ACI-induced $PM_{2.5}$ enhancement approach the more rapid decrease ratio of ARI-induced $PM_{2.5}$ enhancement.

Section 2.2: Could the author also quantitatively list the emissions of primary species related PM$_{2.5}$ formation in 2013 and 2021 so that the readerships have more sense about emission changes in past decade?

Response: Thank you for your comment. The anthropogenic emission data in China for 2013–2021 are obtained from the ABaCAS-EI (Air Benefit and Cost and Attainment Assessment System Emission Inventory) developed by Tsinghua University (Li et al., 2023). Specific emissions of SO$_2$, NO$_x$ (NO and NO$_2$), NH$_3$, PM$_{2.5}$ and VOCs in 2013 and 2021 have been added to the Supplementary Information. Please see Table S2.

Table S2. Regional total emissions of gas and primary particulate matter in China in 2013 and 2021 and its decrease ratio from 2013 to 2021.

|  | SO$_2$ | NO$_x$ | NH$_3$ | PM$_{2.5}$ | VOCs |
|---|---|---|---|---|---|
| 2013 (unit: kt) | 22402 | 27753 | 11040 | 12536 | 25330 |
| 2021 (unit: kt) | 5640 | 15096 | 8847 | 6445 | 22398 |
| (2013-2021)/2013 | 75% | 46% | 20% | 49% | 12% |

Reference

Li SY, Wang SX, Wu QR, Zhang YN, Ouyang DW, Zheng HT, et al. Emission trends of air pollutants and CO$_2$ in China from 2005 to 2021. Earth Syst Sci Data 2023, 15(6): 2279-2294.

Fig. 3: The results are annual mean or for a specific month? Any differences between Jan and July?

Response: Thank you for your comment. The results are the mean for January and July. The results in both January and July show that the percentage decrease in the ARI- and ACI-induced enhancements of sulfate, nitrate, ammonium, and OA is larger than that of OIN and BC (Fig. S10). There is no obvious difference between the two months. Therefore, we discuss the mean results for January and July. We have revised the caption for Fig. 3.

Fig. 3. Percentage decreases (21M13E−21M21E)/21M13E) of the spatial and temporal average ARI- and ACI-induced PM$_{2.5}$ component enhancements in eastern China in January and July caused by the anthropogenic emission reduction from 2013 to 2021.

[Figure]

Fig. S10. Percentage decreases (21M13E−21M21E)/21M13E) of the spatial and temporal averaged ARI- and ACI-induced PM$_{2.5}$ component enhancements in eastern China in January (a) and July (b) caused by emission reduction from 2013 to 2021.

Fig. 5a: Is it the results from all the simulations in all years?

Response: Thank you for your comment. These results are from the simulations for January and July using the background meteorology in 2021 and emissions in 2013, and background meteorology in 2021 and emissions in 2021. The caption of Fig. 5 has been revised.

Fig. 5. (a) The enhanced PM$_{2.5}$ concentrations induced by ARI and ACI at different ambient PM$_{2.5}$ levels. These data are from the simulations for January and July in the experiments of 21M13E and 21M21E. The percentage represents the ratio of the ACI-induced PM$_{2.5}$ enhancement to the sum of ARI- and ACI-induced PM$_{2.5}$ enhancements. (b) The distributions of ambient PM$_{2.5}$ levels in January and July in the experiments of 21M13E and 21M21E.

Line 304: 'lower' -> 'low'

Response: Thank you for your comment. We think "low" is more reasonable; it has been corrected.

Lines 540-544: Does it mean the ACI is always not important over cloud-limited region?

Response: Thank you for your comment. We think that is correct. There is less cloud in the North China Plain in the winter. Given the emissions in 2013 and 2021, the ACI-induced $PM_{2.5}$ enhancements are 0.17 and 0.09 µg m$^{-3}$ in January. These are much smaller than the ACI-induced $PM_{2.5}$ enhancements in the cloudy areas, such as the Sichuan–Chongqing area, which are 7.97 and 4.40 µg m$^{-3}$ in the same time period.

Reply to anonymous referee #2

On behalf of my co-authors, we would like to express our great appreciation for your constructive comments and feedback on our manuscript entitled "Increased Importance of Aerosol-Cloud Interaction for Surface $PM_{2.5}$ Pollution Relative to Aerosol-Radiation Interaction in China with Anthropogenic Emission Reduction" (egusphere-2023-502). We have studied your comments carefully and have revised our manuscript accordingly. Replies to comments are in blue, and the line numbers in the response refer to the clean revised manuscript.

General comments:

The paper titled "Increased Importance of Aerosol-Cloud Interaction for Surface $PM_{2.5}$ Pollution Relative to Aerosol-Radiation Interaction in China With the Anthropogenic Emission Reduction" by Gao et al. investigates changes of the ARI- and ACI-induced $PM_{2.5}$ enhancements for 2013–2021 in China under the background of sharped anthropogenic emission reduction by using WRF-Chem model, and the causes for these changes was further explored from the perspectives of meteorological factors and $PM_{2.5}$ concentration distribution. With this, the authors claimed that the ACI-induced $PM_{2.5}$ enhancement inevitably becomes more important with the decrease of ambient $PM_{2.5}$, that needs to be considered in the formulation of control policies. These results are interesting and useful, although some issues are needed to be clarified and properly improved.

Response: Thank you for your careful evaluation of this manuscript. All your comments and suggestions are important to us. They have direct significance to our paper writing and research work.

Specific and technical comments:

1. Model configuration, I note that the regular chemistry modules like SAPRC-99 and MOSAIC were used in WRF-Chem, with no improvement or revision. Interestingly, the simulated $PM_{2.5}$ was correlated well with the observed values, especially during the heavy polluted period in January, 2013, during which period numerical models usually reported much lower values as suggested by previous studies. Thus, I hope the authors could give the comparison results between the simulated and observed particle chemical components in January 2013, as it had been provided in Fig. S4, which is very important to explore the role of ACI-induced secondary $PM_{2.5}$ component enhancement.

Response: Thank you for your comment. We did try to improve the $PM_{2.5}$ simulation before starting this work. Specifically, we noted the poor ability of nitrate simulation in the WRF-Chem model. Previous studies reported that the nitrate underestimation

might be attributed to the HONO underestimation (Wang et al., 2015; Xue et al., 2020). The source of HONO is originally from some gas-phase chemical reactions in the WRF-Chem, but other HONO sources are lacking, such as hydrolysis of $NO_2$ on humid aerosol surfaces and heterogeneous conversion of $NO_2$ on ground surfaces. It also has been confirmed that these reactions could occur in the atmosphere (Li et al., 2018, Liu et al., 2019). We added these four heterogeneous HONO reactions to the WRF-Chem model (Table S1). The addition of these four reactions enhances atmospheric oxidant and promotes OH radical formation, thereby promoting the formation of gaseous nitric acid through strong OH radicals reacting with $NO_2$. In addition, the reactions of hydrolysis of $NO_2$ on humid aerosol surfaces contributed extra nitric acid. More nitric acid is beneficial for nitrate formation through condensation and heterogeneous reactions. Other specific information can be found in Zhang et al. (2021). This improvement has been added in the revised Supplementary Information. Please see lines 27-43.

Lines 27-43: Before starting this work. We noted the poor ability of nitrate simulation in the WRF-Chem model. Previous studies reported that the nitrate underestimation might be attributed to the HONO underestimation (Wang et al., 2015; Xue et al., 2020). The source of HONO is originally from some gas-phase chemical reactions in the WRF-Chem, but other HONO sources are lacking, such as hydrolysis of $NO_2$ on humid aerosol surfaces and heterogeneous conversion of $NO_2$ on ground surfaces. It also has been confirmed that these reactions could occur in the atmosphere (Li et al., 2018, Liu et al., 2019). We added these four heterogeneous HONO reactions to the WRF-Chem model (Table S1). The addition of these four reactions enhances atmospheric oxidant and promotes OH radical formation, thereby promoting the formation of gaseous nitric acid through strong OH radicals reacting with $NO_2$. In addition, the reactions of hydrolysis of $NO_2$ on humid aerosol surfaces contributed extra gaseous nitric acid. More gaseous nitric acid is beneficial for nitrate formation through condensation and heterogeneous reactions. Other specific information can be found in Zhang et al. (2021).

Table S1. Heterogeneous HONO reactions added in WRF-Chem (from Zhang et al., 2021).

| number | Reaction | Reference |
|---|---|---|
| (1) | $NO_2 + aerosol = 0.5 \times HONO + 0.5 \times HNO_3$ | Liu et al., (2019) |
| (2) | $NO_2 + aerosol = 0.5 \times HONO + 0.5 \times HNO_3$ | Liu et al., (2019) |
| (3) | $NO_2 + ground = HONO$ | Li et al., (2018), Liu et al., (2019) |
| (4) | $NO_2 + ground + h\nu = HONO$ | Liu et al., (2019) |

As reviewer mentioned, we reproduce the $PM_{2.5}$ pollution in January 2013 in China accurately. The $PM_{2.5}$ components data in January 2013 are very rare, and we sourced three sets of data, in Beijing, Handan, and Shanghai. As shown in Fig. S4, we find that the simulated $PM_{2.5}$ components in January 2013 are reproduced well generally,

but they still have small discrepancies compared with the observations in three cities. Specifically, the simulated PM$_{2.5}$ components are larger than half of observational PM$_{2.5}$ components and less than the double observational PM$_{2.5}$ components (Fig. S4).

We also compared our simulated results with previous studies. We find that some previous studies have also successfully reproduced the PM$_{2.5}$ pollution in January 2013, but they also have certain discrepancies in the PM$_{2.5}$ components (Li et al., 2015; Mattias et al., 2017). Taking modeling in Beijing and Shanghai as two examples, Mattias et al. (2017) found that their model (COSMO-CLM-CMAQ) is generally able to reproduce the high PM$_{2.5}$ level measured in situ close to the ground in Beijing, but it overestimates the BC concentration. Li et al., (2015) reported that the simulated PM$_{2.5}$ concentration is slightly overestimated owing to the overestimation of fugitive dust. Therefore, the models all have difficulty in reproducing all the PM$_{2.5}$ chemical components very accurately. For 2013, our simulated PM$_{2.5}$ concentrations in January and July are good and simulated PM$_{2.5}$ components in January have small discrepancies with the observations (Fig. S4). The evaluation of the simulation in January 2013 has been added to the revised manuscript. Please see lines 285-291.

Lines 285-291: Given that the PM$_{2.5}$ components data in 2013 are very rare, we sourced three sets of data in January 2013, respectively in Beijing (Mattias et al., 2017), Handan (Zhang et al., 2015), and Shanghai (Li et al., 2015). The results show that the simulated PM$_{2.5}$ components are reproduced well generally. Specifically, the simulated PM$_{2.5}$ components are larger than half of observational PM$_{2.5}$ components and less than the double observational PM$_{2.5}$ components (Fig. S4).

[Figure]

[Figure]

[Figure]

Fig. S4. Comparisons between PM$_{2.5}$ component observations and simulations in Handan, Beijing and Shanghai in January 2013. The blue points represent double or half of the PM$_{2.5}$ component observations. OA and BC represent organic aerosol and black carbon.

2. Line 274-276,it seems that not only sulfate, but the majority of BC and parts of OC are beyond the ratios of 2.0 in January 2021 (as showed in Fig. S4). I do not think such discrepancies is minor and will not cause obvious uncertainty in this research. The simulated BC alone would cause high uncertainty in the ARI-induced PM$_{2.5}$ enhancements. In addition, the higher values in the simulated OC is also weird. Please added more explanation here.

Response: Thank you for your comment. To test the impact of simulated BC overestimation in January 2021 on ARI-induced PM$_{2.5}$ enhancement, we utilize another set of particulate matter (PM) source profiles (Liu et al., 2018) and conduct the simulations for January 2021. The results indicate that the ratios of simulated BC concentration to observational BC concentration are within 2.0. The ARI-induced PM$_{2.5}$ enhancement is 1.33 µg m$^{-3}$, which shows a negligible difference from the result (1.37 µg m$^{-3}$) obtained using original PM source profiles (Fig. S6). However, the reduction in simulated BC concentration in January 2021 does not necessarily mean that this set of PM source profiles is better than the original PM source profiles, because this might be an accidental result caused by other uncertainties. For example, the current model underestimates the wet deposition of BC due to neglecting the increase in BC hygroscopicity brought about by BC aging. If this process is considered in the model, simulated BC concentrations might be better reproduced using original PM source profiles. Therefore, in this study, we still use the original results for our analysis.

[Figure]

Fig. S6. The ratios of simulation to observation of BC and ARI-induced PM$_{2.5}$ enhancement (unit: µg m$^{-3}$) obtained from original PM source profiles and another set of PM source profiles in January 2021.

The reviewer also identified that the model underestimates the sulfate concentration and overestimates the part of OC concentration in January 2021. We think that neither of these discrepancies will cause significant uncertainties in ARI- and ACI-induced PM$_{2.5}$ enhancement. Specifically, the majority of aerosol is scattering aerosol and the PM$_{2.5}$ concentration in January 2021 is reproduced well. Therefore, we think that the impact of the sulfate underestimation on the ARI-induced PM$_{2.5}$ enhancement would be largely offset by the overestimation of other scattering aerosol components, such as OC. In addition, the OC overestimation should not bring significant uncertainty to ACI-induced PM$_{2.5}$ enhancement either, because of the relatively lower hygroscopicity of OC compared to secondary inorganic aerosol. The underestimation of sulfate simulation in January 2021 also minimally affects ACI-induced PM$_{2.5}$ enhancement because the sulfate underestimation mainly occurs in the North China Plain, where cloud cover is low. In contrast, in southern cities such as Mianyang city in Sichuang province where there is plenty of cloud cover, the sulfate simulation was 4.19 µg m$^{-3}$ in January 2021, which is very close to the observed value of 4.25 µg m$^{-3}$ (Lin et al., 2022).

Corresponding discussions have been added in lines 292-332.

Lines 292-332: Fig. S5 shows the ratios of observation to simulation of ammonium, sulfate, BC and organic carbon (OC) in January and July 2021. The results exhibit that almost all the ratios of PM$_{2.5}$ components are located between 0.5 and 2.0, while some ratios of sulfate in January, part of OC in January, and BC in January and July

are beyond this range. But these discrepancies will not cause obvious uncertainties in this research. Specifically, considering BC low hygroscopicity, BC overestimations in January and July 2021 probably bring low uncertainties in ACI-induced PM$_{2.5}$ enhancement. To test the impact of simulated BC overestimation in January 2021 on ARI-induced PM$_{2.5}$ enhancement, we utilize another set of particulate matter (PM) source profiles (Liu et al., 2018) and conduct the simulations for January 2021. The results indicate that the ratios of simulated BC concentration to observational BC concentration are within 2.0. The ARI-induced PM$_{2.5}$ enhancement is 1.33 µg m$^{-3}$, which shows a negligible difference from the result (1.37 µg m$^{-3}$) obtained using original PM source profiles (Fig. S6). In view of the results in January 2021, the BC overestimation in July 2021 also probably brings low uncertainties in ARI-induced PM$_{2.5}$ enhancement. However, the reduction in simulated BC concentration in January 2021 does not necessarily mean that this set of PM source profiles is better than the original PM source profiles, because this might be an accidental result caused by other uncertainties. For example, the current model underestimates the wet deposition of BC due to neglecting the increase in BC hygroscopicity brought about by BC aging. If this process is considered in the model, simulated BC concentrations might be better reproduced using original PM source profiles. Therefore, in this study, we still use the original results for our analysis. The model also underestimates the sulfate concentration and overestimates the part of OC concentration in January 2021. We think that neither of these discrepancies will cause significant uncertainties in ARI- and ACI-induced PM$_{2.5}$ enhancement. Specifically, the majority of aerosol is scattering aerosol and the PM$_{2.5}$ concentration in January 2021 is reproduced well. Therefore, we think that the impact of the sulfate underestimation on the ARI-induced PM$_{2.5}$ enhancement would be largely offset by the overestimation of other scattering aerosol components, such as OC. In addition, the OC overestimation should not bring significant uncertainty to ACI-induced PM$_{2.5}$ enhancement either, because of the relatively lower hygroscopicity of OC compared to secondary inorganic aerosol. The underestimation of sulfate simulation in January 2021 also minimally affects ACI-induced PM$_{2.5}$ enhancement because the sulfate underestimation mainly occurs in the North China Plain, where cloud cover is low. In contrast, in southern cities such as Mianyang city in Sichuang province where there is plenty of cloud cover, the sulfate simulation was 4.19 µg m$^{-3}$ in January 2021, which is very close to the observed value of 4.25 µg m$^{-3}$ (Lin et al., 2022).

3. Line 432-435, it is unclear about the high ambient PM$_{2.5}$ concentration conditions, is it about the January 2021 or July 2021? If it is about the latter, please provide more discussion about the relationship of PM and CCN to show the CCN is enough.

Response: Thank you for your comment. The description of the high ambient PM$_{2.5}$ concentration condition originally represents the regional and temporal average PM$_{2.5}$ concentration in eastern China in January and July, which is simulated using background meteorology in 2021 and emissions in 2013.

The analysis of the relationship between $PM_{2.5}$ and CCN might not accurately indicate whether there is an adequate amount of CCN because CCN is a diagnostic variable in the model and varies continuously with the change in $PM_{2.5}$ concentration without being constrained by cloud water. Here, to better elucidate the relationship between the change in $PM_{2.5}$ concentration and the change in ACI-induced SWDOWN reduction, we find it more straightforward to discuss the impact of $PM_{2.5}$ reduction on the change in LWP and cloud effective radius (Re) induced by ACI. In our simulations, the influence of ACI-induced Re change is relatively smaller than that of ACI-induced LWP change with a large decrease in $PM_{2.5}$ concentration (Fig. S7). Therefore, we are only concerned with change in ACI-induced LWP with a reduction in $PM_{2.5}$. As shown in Fig. S8a, when the ambient $PM_{2.5}$ concentration exceeded 15 $\mu g\ m^{-3}$, the decrease in ACI-induced LWP increase is relatively low with a $PM_{2.5}$ reduction from 120 to 15 $\mu g\ m^{-3}$, indicating that aerosols are not a key limiting factor to cloud formation in this range. In contrast, when the ambient $PM_{2.5}$ concentration decreases from 15 to 0 $\mu g\ m^{-3}$, the decrease in ACI-induced LWP increase is much faster. Previous studies have also demonstrated that the decrease in ACI-induced LWP increase is relatively fast or slow with the ambient $PM_{2.5}$ reduction in the $PM_{2.5}$-clean or polluted condition, respectively (Myhre et al., 2007; Savane et al., 2015).

The regional and temporal average $PM_{2.5}$ concentration in eastern China in January and July simulated using background meteorology in 2021 and emissions in 2013 is 63 and 25 $\mu g\ m^{-3}$, which is much higher than 15 $\mu g\ m^{-3}$. Therefore, the decrease in ACI-induced SWDOWN reduction in both months is weak.

We have revised the original descriptions. Please see lines 489-493 in the revised manuscript.

Lines 489-493: The reason is that the decrease in ambient $PM_{2.5}$ concentration directly weakens the ARI-induced SWDOWN reduction, but it has only a minor impact on the ACI-induced SWDOWN reduction because the change in LWP and cloud effective radius (Re) induced by ACI is not sensitive to $PM_{2.5}$ reduction in the $PM_{2.5}$-polluted regime (Fig. S8a).

[Figure]

Fig. S7. Change in aerosol-induced LWP (unit: g m$^{-2}$), Re (unit: μm) and ACI-induced SWDOWN reduction (unit: W m$^{-2}$) in January owing to the emission reduction from 2013 to 2021.

[Figure]

Fig. S8. Change in LWP and SWDOWN induced by ACI at different PM$_{2.5}$ levels.

4. Line 477-484, the explanation about the ARI- and ACI-induced PM$_{2.5}$ enhancements from the perspective of PM$_{2.5}$ concentration is weak, and no more new information was deduced here as illustrated in section 3.4.1. I would suggest to add more discussion on the meteorological conditions and PM2,5 chemical compositions under different PM pollution levels, to explain the changes of ARI- and ACI-induced PM$_{2.5}$ enhancements.

Response: Thank you for your comment. We try to analyze the change in ARI- and ACI-induced PM$_{2.5}$ enhancement by discussing the change in ARI- and ACI-induced meteorological conditions and PM$_{2.5}$ chemical components. The changes in ARI- and ACI-induced PM$_{2.5}$ components between the different pollution levels could not be directly compared. As an example to demonstrate this process, if the PM$_{2.5}$ concentration in Shanghai and Lanzhou is 80 and 100 µg m$^{-3}$, when we discuss the changes in ARI- and ACI-induced PM$_{2.5}$ components with PM$_{2.5}$ concentration decreasing from 100 to 80 µg m$^{-3}$, we would probably find ARI- and ACI-induced OIN enhancement mainly decrease owing to large and small contribution of dust to

PM$_{2.5}$ concentration in Lanzhou and Shanghai. Therefore, we do not think that the ARI- and ACI-induced PM$_{2.5}$ components between the different PM$_{2.5}$ pollution levels are comparable.

We focus on the relationship between the change in ARI and ACI-induced meteorological conditions and change in ARI- and ACI-induced PM$_{2.5}$ enhancement. We find that, in the heavily PM$_{2.5}$-polluted regime (135–180 μg m$^{-3}$), the decrease in SWDOWN induced by ARI is much larger than that induced by ACI (Fig. S9a). Then, the decrease in PBLH and T2 and the increase in RH induced by ARI are also larger than those induced by ACI (Fig. S9b–d). Thus, the enhanced PM$_{2.5}$ induced by the ARI is much larger than that by the ACI (Fig. 5a). However, when the PM$_{2.5}$ concentration decrease to the range of 15–45 μg m$^{-3}$, the decrease in SWDOWN, PBLH, and T2 and the increase in RH induced by ACI significantly exceed those induced by ARI. Thus, the ACI-induced PM$_{2.5}$ enhancement significantly exceeds the ARI-induced PM$_{2.5}$ enhancement and becomes more important. This indicates the fast decrease in the ARI-induced PM$_{2.5}$ enhancement and the increasing contribution of the ACI-induced PM$_{2.5}$ enhancement with the decrease in the PM$_{2.5}$ concentration. In summary, the percentage decrease in the PM$_{2.5}$ enhancement induced by ACI is weaker than that induced by ARI with the decrease of PM$_{2.5}$ concentration because of the lower percentage decrease in the ACI-induced SWDOWN, which causes the lower percentage decrease in the ACI-induced PBLH and T2 reduction and the RH increase. Furthermore, as shown in Fig. S8a, the low percentage decrease in the ACI-induced SWDOWN reduction is due to a low decrease in the ACI-induced LWP in the PM$_{2.5}$-polluted regime.

Our findings thus indicate that it is not an accident that the increasing importance of ACI-induced PM$_{2.5}$ enhancement occurs in China with emission reduction from 2013 to 2021, and it is a matter of course with ambient PM$_{2.5}$ decrease. The change in ARI- and ACI-induced meteorological factors is a major reason for the change in ARI- and ACI-induced PM$_{2.5}$ enhancement.

We have revised the original description. Please see lines 541-560 in the revised manuscript.

Lines 541-560: As shown in Fig. 5a, in the heavily PM$_{2.5}$-polluted regime (135–180 μg m$^{-3}$), the decrease in SWDOWN induced by ARI is much larger than that induced by ACI (Fig. S9a). Then, the decrease in PBLH and T2 and the increase in RH induced by ARI are also larger than those induced by ACI (Fig. S9b–d). Thus, the enhanced PM$_{2.5}$ induced by the ARI is much larger than that by the ACI (Fig. 5a). However, when the PM$_{2.5}$ concentration decrease to the range of 15–45 μg m$^{-3}$, the decrease in SWDOWN, PBLH, and T2 and the increase in RH induced by ACI significantly exceed those induced by ARI. Thus, the ACI-induced PM$_{2.5}$ enhancement significantly exceeds the ARI-induced PM$_{2.5}$ enhancement and becomes more important. This indicates the fast decrease in the ARI-induced PM$_{2.5}$ enhancement and the increasing contribution of the ACI-induced PM$_{2.5}$ enhancement

with the decrease in the PM$_{2.5}$ concentration. In summary, the percentage decrease in the PM$_{2.5}$ enhancement induced by ACI is weaker than that induced by ARI with the decrease of PM$_{2.5}$ concentration because of the lower percentage decrease in the ACI-induced SWDOWN, which causes the lower percentage decrease in the ACI-induced PBLH and T2 reduction and the RH increase. Furthermore, as shown in Fig. S8a, the low percentage decrease in the ACI-induced SWDOWN reduction is due to a low decrease in the ACI-induced LWP in the PM$_{2.5}$-polluted regime.

[Figure]

Fig. 5. (a) The enhanced PM$_{2.5}$ concentrations induced by ARI and ACI at different ambient PM$_{2.5}$ levels. These data are from the simulations for January and July in the experiments of 21M13E and 21M21E. The percentage represents the ratio of the ACI-induced PM$_{2.5}$ enhancement to the sum of ARI- and ACI-induced PM$_{2.5}$ enhancements.

[Figure]

[Figure]

[Figure]

[Figure]

Fig. S9. Change in SWDOWN (a), PBLH (b), T2 (c), and RH (d) induced by ARI and ACI at different ambient $PM_{2.5}$ levels. These data are from the simulations for January and July in the experiments of 21M13E and 21M21E.

Minor revisions:

1. Line 109-114, about the conclusions cited from Moch et al. (2022) and Zhang et al. (2022), more details like which regions and which periods should be provided.

Response: Thank you for your comment. Detailed information has been added to the conclusions. Please see lines 111-117 in the revised manuscript.

Moch et al. (2022) found that the decrease in mean $PM_{2.5}$ concentration from the winter months of 2012–2013 to the winter months of 2016–2017 in China weakened the cloud–snowfall–albedo feedback induced by the aerosol semi-direct effect. For air quality, Zhang et al. (2022) found that the decrease in black carbon from 2013 to 2017 in China reduced the enhanced $PM_{2.5}$ concentration induced by the ARI by 1.8 $\mu g \, m^{-3}$ in January and 0.3 $\mu g \, m^{-3}$ in July.

2. Line 176-177, about the influence of prognostic aerosol in the model, more explanation is better to provided.

Response: Thank you for your comment. The prognostic aerosol can only be activated as CCN. It does not directly contribute to ice nucleation, which is only influenced by air temperature and supersaturation (Kanji et al., 2017). Furthermore, CCN would influence grid-scale clouds. However, limited by the horizontal resolution of 27 km × 27 km, cumulus clouds could not be resolved in this grid. Therefore, prognostic

aerosol does not influence ice nucleation and cumulus clouds in the model. This explanation has been added to the revised manuscript. Please see lines 186-191

3. Line 388-389, please define the inorganic aerosol (OIN) here, as sulfate and nitrate is also recognized as inorganic aerosol.

Response: Thank you for your comment. Other organic aerosol (OIN) refers to inorganic compositions other than sulfate, nitrate, ammonium, and BC. These compositions include sea salt and mineral elements. This has been corrected in the revised manuscript. Please see lines 444-446.

References

Kanji, Z. A., Ladino, L. A., Wex, H., Boose, Y., Burkert-Kohn, M., Cziczo, D. J., and Kramer, M.: Overview of Ice Nucleating Particles, Meteor Mon, 58, 10.1175/Amsmonographs-D-16-0006.1, 2017.

Li, L., An, J. Y., Zhou, M., Yan, R. S., Huang, C., Lu, Q., Lin, L., Wang, Y. J., Tao, S. K., Qiao, L. P., Zhu, S. H., and Chen, C. H.: Source apportionment of fine particles and its chemical components over the Yangtze River Delta, China during a heavy haze pollution episode, Atmos Environ, 123, 415-429, 10.1016/j.atmosenv.2015.06.051, 2015.

Li, D. D., Xue, L. K., Wen, L., Wang, X. F., Chen, T. S., Mellouki, A., Chen, J. M., and Wang, W. X.: Characteristics and sources of nitrous acid in an urban atmosphere of northern China: Results from 1-yr continuous observations, Atmos Environ, 182, 296-306, 10.1016/j.atmosenv.2018.03.033, 2018.

Lin, C. J.: Characteristics and Sources of Water-soluble Inorganic Ions in Atmospheric Particulate Matter and Rainfall in the suburb of Mianyang, Master, Southwest University of Science and Technology, 2022.

Liu, Y. Y., Xing, J., Wang, S. X., Fu, X., and Zheng, H. T.: Source-specific speciation profiles of PM2.5 for heavy metals and their anthropogenic emissions in China, Environ Pollut, 239, 544-553, 10.1016/j.envpol.2018.04.047, 2018.

Liu, Y. H., Lu, K. D., Li, X., Dong, H. B., Tan, Z. F., Wang, H. C., Zou, Q., Wu, Y. S., Zeng, L. M., Hu, M., Min, K. E., Kecorius, S., Wiedensohler, A., and Zhang, Y. H.: A Comprehensive Model Test of the HONO Sources Constrained to Field Measurements at Rural North China Plain, Environ Sci Technol, 53, 3517-3525, 10.1021/acs.est.8b06367, 2019.

Matthias, V., Aulinger, A., Bieser, J., Chen, Y. J., Geyer, B., Gao, J., Quante, M., and Zhang, F.: Modeling high aerosol loads in China in January 2013, Urban Clim, 22, 35-50, 10.1016/j.uclim.2016.04.005, 2017.

Moch, J. M., Mickley, L. J., Keller, C. A., Bian, H. S., Lundgren, E. W., Zhai, S. X., and Jacob, D. J.: Aerosol-Radiation Interactions in China in Winter: Competing

Effects of Reduced Shortwave Radiation and Cloud-Snowfall-Albedo Feedbacks Under Rapidly Changing Emissions, J Geophys Res-Atmos, 127, ARTN e2021JD035442

10.1029/2021JD035442, 2022.

Myhre, G., Stordal, F., Johnsrud, M., Kaufman, Y. J., Rosenfeld, D., Storelvmo, T., Kristjansson, J. E., Berntsen, T. K., Myhre, A., and Isaksen, I. S. A.: Aerosol-cloud interaction inferred from MODIS satellite data and global aerosol models, Atmos Chem Phys, 7, 3081-3101, DOI 10.5194/acp-7-3081-2007, 2007.

Savane, O. S., Vant-Hull, B., Mahani, S., and Khanbilvardi, R.: Effects of Aerosol on Cloud Liquid Water Path: Statistical Method a Potential Source for Divergence in Past Observation Based Correlative Studies, Atmosphere-Basel, 6, 273-298, 10.3390/atmos6030273, 2015.

Wang, L. W., Wen, L., Xu, C. H., Chen, J. M., Wang, X. F., Yang, L. X., Wang, W. X., Yang, X., Sui, X., Yao, L., and Zhang, Q. Z.: HONO and its potential source particulate nitrite at an urban site in North China during the cold season, Sci Total Environ, 538, 93-101, 10.1016/j.scitotenv.2015.08.032, 2015.

Xue, C. Y., Zhang, C. L., Ye, C., Liu, P. F., Catoire, V., Krysztofiak, G., Chen, H., Ren, Y. G., Zhao, X. X., Wang, J. H., Zhang, F., Zhang, C. X., Zhang, J. W., An, J. L., Wang, T., Chen, J. M., Kleffmann, J., Mellouki, A., and Mu, Y. J.: HONO Budget and Its Role in Nitrate Formation in the Rural North China Plain, Environ Sci Technol, 54, 11048-11057, 10.1021/acs.est.0c01832, 2020.

Zhang, F. F.: Characteristics of Air Pollution and Chemical Composition of PM2.5 in Handan Master, College of Urban Construction, Hebei University of Engineering, 2015.

Zhang, S. P., Sarwar, G., Xing, J., Chu, B. W., Xue, C. Y., Sarav, A., Ding, D. A., Zheng, H. T., Mu, Y. J., Duan, F. K., Ma, T., and He, H.: Improving the representation of HONO chemistry in CMAQ and examining its impact on haze over China, Atmos Chem Phys, 21, 15809-15826, 10.5194/acp-21-15809-2021, 2021.

Zhang, F. F., Xing, J., Ding, D. A., Wang, J. D., Zheng, H. T., Zhao, B., Qi, L., and Wang, S. X.: Role of black carbon in modulating aerosol direct effects driven by air pollution controls during 2013-2017 in China, Sci Total Environ, 832, ARTN 154928

10.1016/j.scitotenv.2022.154928, 2022.

---

## Author Response (AR3)

Thanks for addressing the referee comments. Both reviewers recommend publication, and I am happy to accept your manuscript for publication. Please consider the suggestion to use different colors for the arrows in Fig. 6, and please add a graphical abstract (will be required for final publication in ACP).

Response: Thank you for your comments. We have made revisions to Fig. 6 by using grey as the color of the text box and orange for the arrows inside the text boxes. This change enhances the clarity of the arrows. Additionally, we can include Fig. 6 as a graphical abstract.